# Chaperone mediated detection of small molecule target binding in cells

Kelvin F. Cho[1], Taylur P. Ma[2], Christopher M. Rose [2], Donald S. Kirkpatrick [2], Kebing Yu[2] & Robert A. Blake[1]*

The ability to quantitatively measure a small molecule's interactions with its protein target(s) is crucial for both mechanistic studies of signaling pathways and in drug discovery. However, current methods to achieve this have specific requirements that can limit their application or interpretation. Here we describe a complementary target-engagement method, HIPStA (Heat Shock Protein Inhibition Protein Stability Assay), a high-throughput method to assess small molecule binding to endogenous, unmodified target protein(s) in cells. The methodology relies on the change in protein turnover when chaperones, such as HSP90, are inhibited and the stabilization effect that drug-target binding has on this change. We use HIPStA to measure drug binding to three different classes of drug targets (receptor tyrosine kinases, nuclear hormone receptors, and cytoplasmic protein kinases), via quantitative fluorescence imaging. We further demonstrate its utility by pairing the method with quantitative mass spectrometry to identify previously unknown targets of a receptor tyrosine kinase inhibitor.

---

[1] Department of Biochemical and Cellular Pharmacology, Genentech Inc., South San Francisco, CA 94080, USA. [2] Department of Microchemistry, Proteomics & Lipidomics, Genentech Inc., South San Francisco, CA 94080, USA. *email: blake.robert@gene.com

Small molecules are frequently used as tools in biological research for elucidating mechanisms of cellular signaling; thus, it is critical to understand the nature and specificity of the target proteins they bind to and modulate. While in vitro biochemical measurements provide important information, they do not always recapitulate cellular physiology[1]. Several methods have been developed to address cellular target engagement, but each has its respective requirements and applications. The Cellular Thermal Stability Assay (CETSA®) was designed to measure drug-target engagement in cells, based on the principle that a protein's thermodynamic stability is affected upon ligand binding[2,3]. Classic CETSA® involves treating cells with compounds, heating the cells to different temperatures, lysis, and subsequent analysis by Western blot. While several groups have adapted CETSA® for high-throughput screening and profiling applications, these approaches either require developing specialized reagents for each target of interest[4,5] or require tagging of the protein[6]. Furthermore, all CETSA® and CETSA®-related approaches require cells to be heated, which can be highly perturbative. Alternative approaches to measuring cellular target engagement have other requirements: FLIM (Fluorescence Lifetime Imaging Microscopy)-based approaches require fluorophore-conjugated ligands[7,8], while nanoBRET, InCell Hunter™, dynamic proteomics, and LUMIER-BACON-based methods require tagged proteins[9–12].

To provide a complementary target engagement technique, we here develop the Heat Shock Protein Inhibition Protein Stability Assay (HIPStA) as a high-throughput method to measure small molecule binding to endogenous targets in cells. The foundation of the HIPStA method is that many proteins are stabilized by the chaperone protein Heat Shock Protein 90 (HSP90)[13]. HSP90 is expressed at physiological temperatures and is essential for maintaining protein stability by refolding partially unfolded proteins back into their correct conformations[13,14]. Based on the same principle that a protein's thermodynamic stability is affected upon ligand binding, HIPStA employs a pharmacological inhibitor of HSP90 to induce protein instability, instead of using heat to destabilize proteins, as in CETSA®. We use HIPStA to show that the pharmacological inhibition of HSP90 could be leveraged to measure small molecule target binding to various drug targets using quantitative fluorescence imaging. We then pair HIPStA with quantitative mass spectrometry to globally profile the cellular proteome after drug treatment, identifying previously unknown targets of small molecule inhibitors.

## Results

### HSP90 inhibition to measure drug-target binding to ERα.
In order to test the ability of the HIPStA method to measure drug-target interactions, we selected three drug targets with distinct subcellular localization profiles: ERα (nuclear hormone receptor), c-RAF (cytoplasmic protein kinase), and HER2 (receptor tyrosine kinase)[15–17]. We first tested ERα, one of several nuclear hormone receptors that have been shown to be HSP90 clients[18]. To observe the effects of HSP90 inhibition on ERα, we treated MCF-7 breast cancer cells with 17-(allylamino)-17-demethoxygeldanamycin (17-AAG), a widely utilized and well characterized HSP90 inhibitor[19]. Using immunofluorescence imaging, we observed a reduction of ERα upon treatment with 17-AAG, confirming that inhibition of HSP90 can destabilize ERα (Fig. 1a). This reduction was both concentration-dependent, with a half maximal effective concentration ($EC_{50}$) of 25 nM 17-AAG (Fig. 1b), and time-dependent, with a half-life of 428 min (Supplementary Fig. 1A). Pre-treatment with a well-known ERα ligand, Endoxifen[20] (Fig. 1c), inhibited the 17-AAG induced reduction in ERα protein levels, reflected in an elevation of the plateau level in the destabilization curves (Fig. 1d).

This suggests an increasing population of ERα protein that is resistant to destabilization in the presence of increasing concentrations of Endoxifen. We confirmed that this effect of Endoxifen on ERα was not due to an indirect effect on ERα (ESR1) mRNA transcript levels (Supplementary Fig. 2A). We further confirmed that this effect was independent of de novo protein synthesis, since repeating the experiment with cycloheximide co-treatment showed no difference in the changes of ERα levels (Supplementary Fig. 2B). The observation that small molecules targeting the protein of interest elevate the plateau of the 17-AAG-induced destabilization curve is of significant practical advantage. Instead of having to carry out a full titration series of 17-AAG, a single maximally effective concentration of 17-AAG and titration of test compound are sufficient to elucidate the stabilizing effect of the small molecule (Fig. 1e), greatly improving throughput. By measuring the proportion of ERα stabilized at each concentration of Endoxifen, we derived a HIPStA $EC_{50}$ value of 0.16 nM (Fig. 1e). This is within 10-fold of published cellular $EC_{50}$ values for Endoxifen, which range between 1-10 nM[20–22].

### HIPStA extended to measure drug-target binding to c-RAF.
Having demonstrated proof of concept for using HIPStA to measure drug-target binding to ERα, we wanted to test its generalizability to other HSP90 clients. We applied HIPStA to the cytoplasmic protein kinase, c-RAF, which has been reported to be an HSP90 client[23]. Comparing c-RAF immunofluorescence images in 17-AAG-treated A549 lung cancer cells to DMSO-treated controls; we detected a reduction in c-RAF protein only in the cells with the highest c-RAF expression (Fig. 2a). Despite this, we were able to measure an $EC_{50}$ of 54 nM for the effect of 17-AAG on c-RAF (Fig. 2b). To test the effects of drug binding on c-RAF stabilization, we selected two c-RAF inhibitors, BGB-283 (enantiomer) and a pan-RAF inhibitor[24,25] (Fig. 2c, d). We observed examples of drugs causing a direct effect on protein abundance, elevating c-RAF levels at high concentrations, even in the absence of HSP90 inhibition (Fig. 2e, f). This result illustrates and confirms observations from dynamic proteomics studies[12] that a drug-target interaction can sometimes be detected without having to resort to a protein destabilization strategy. While we still observed elevation in the plateaus of the destabilization curves upon dual-treatment (Fig. 2g, h), we could not fully attribute these effects to the compounds preventing 17-AAG destabilization specifically, given the compounds' direct effects on c-RAF levels.

### HIPStA extended to measure drug-target binding to HER2.
Next, we applied HIPStA to the receptor tyrosine kinase, HER2, another reported HSP90 client[26]. We observed that 17-AAG caused a marked reduction in the level of HER2 protein in MCF-7 neo-HER2 cells (Fig. 3a). This reduction was concentration-dependent, with a half maximal effective concentration ($EC_{50}$) of 91 nM (Fig. 3b), and a half-life >500 min (Supplementary Fig. 1B). We then examined the effect of known HER2 inhibitors on 17-AAG-induced destabilization of HER2: Lapatinib, HY-14674, and TAK285[27–29] (Fig. 3c). Addition of small molecules targeting HER2 rescued the 17-AAG-induced reduction of HER2 protein, seen as an elevation of the plateau for all three sets of destabilization curves (Fig. 3d). We observed similar results in the SK-BR-3 and AU565 cell lines, and using the alternative HSP90 inhibitor, XL888 (Supplementary Fig. 3A-E). This result is consistent with there being a population of HER2 that becomes resistant to destabilization in the presence of increasing concentrations of HER2 inhibitors. A single maximally effective concentration of 17-AAG and titration of the test compounds were sufficient to detect the stabilizing effects (Fig. 3e). By measuring the proportion of HER2 stabilized at each

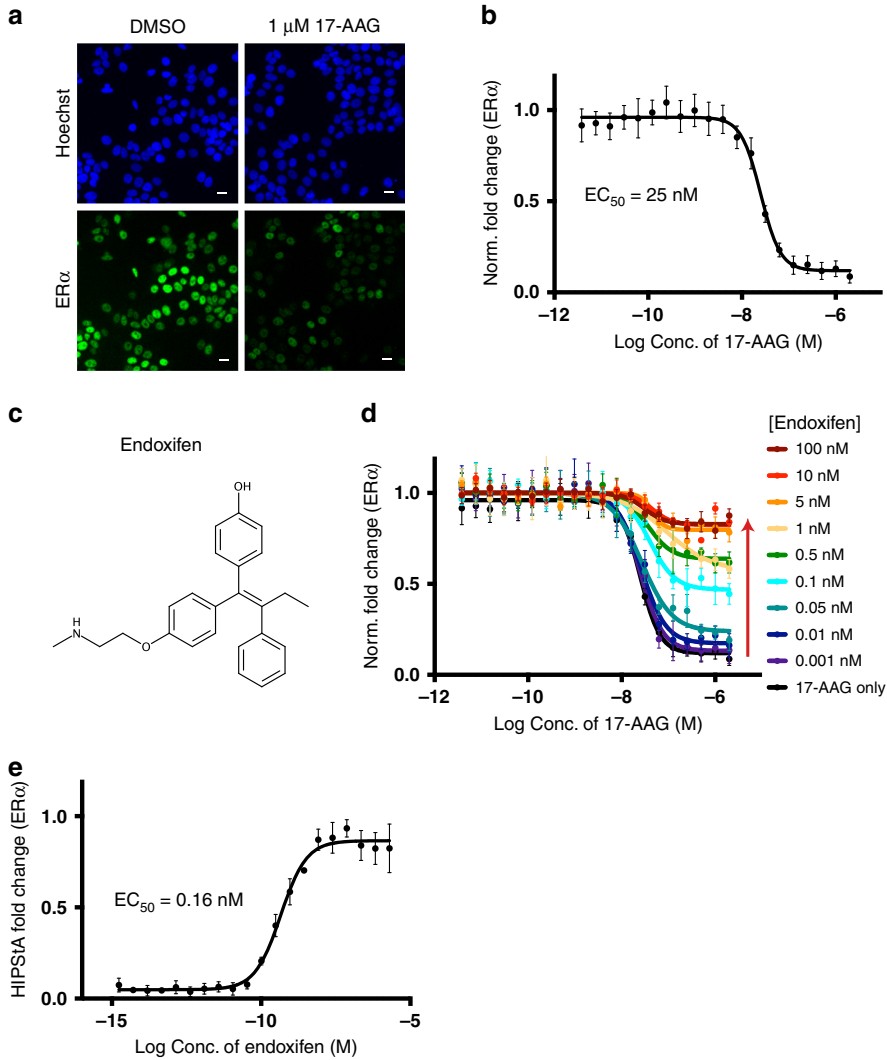

**Fig. 1 Quantitative fluorescence imaging measuring the effect of 17-AAG on ERα. a** Immunofluorescence images of ERα protein in MCF-7 cells treated with DMSO or 17-AAG (1 μM) for 4 h. The upper panels show Hoechst 33342 staining of the DNA (blue) indicating the nuclear region, while the lower panels show the ERα immunofluorescence stain. Scale bar −20 μm. **b** The change in mean ERα protein per cell (MCF-7) was calculated for a range of concentrations of 17-AAG and expressed relative to the ERα level in DMSO treated samples and the no primary antibody control sample. **c** Endoxifen chemical structure. **d** Measurement of the relative ERα protein levels in MCF-7 cells pre-treated with a range of concentrations of Endoxifen then treated with a titration of 17-AAG (x-axis) for 4 h. **e** Measurement of the relative ERα protein levels in MCF-7 cells pre-treated with a titration of Endoxifen then treated with 2 μM 17-AAG. ERα protein levels are expressed as the fraction of ERα present in the co-treatment condition relative to the corresponding concentration of inhibitor only (HIPStA Fold Change). The relative $EC_{50}$ value was calculated using a four parameter quadratic curve fit in PRISM. Error bars represent standard error of mean ($n = 3$). Source data are provided as a Source Data file.

concentration, we derived HIPStA $EC_{50}$ values for HER2 protein stabilization by each compound: The Lapatinib HIPStA $EC_{50}$ was 1.1 nM (the published cellular $IC_{50}$ values for Lapatinib range between 0.025 and 1.2 μM[30–32] and the HER2 biochemical $IC_{50}$ is 25 nM; Supplementary Table 1). The HY-14674 HIPStA $EC_{50}$ was 77 nM (the HER2 biochemical $IC_{50}$ is 18 nM; Supplementary Table 1). The TAK-285 HIPStA $EC_{50}$ was 79 nM (the published cellular TAK-285 is 17 nM[33,34] and we measured a biochemical $IC_{50}$ of 43 nM; Supplementary Table 1 and Fig. 3e).

**HIPStA global proteome profiling to identify drug targets.** Having demonstrated that HIPStA can detect cellular small molecule target interactions for three distinct classes of proteins, we next investigated whether the method could be extended to identify novel small molecule-protein interactions. To address

this at the proteome scale, we combined HIPStA with isobaric labeling and quantitative mass spectrometry (Fig. 4a) to study the HER2 inhibitor TAK-285. Independent experiments were performed testing the effects of TAK-285 at 100 nM and 1 μM, respectively. In each case, TAK-285 was tested either alone or in combination with 1 μM 17-AAG. The corresponding control, 1 μM 17-AAG alone, revealed the HSP90 dependent fraction of the proteome and provided empirical data on the range of target proteins accessible to the HIPStA method. 17-AAG treatment resulted in the down-regulation of 122 proteins after 6 h and 179 proteins after 8 h (>25% decrease, $p < 0.05$ (two-tailed student $t$-test); Fig. 4b, Supplementary Data 1 and Supplementary Fig. 4A). Consistent with reports that many kinases are HSP90 clients[35], the density plot of 229 quantified kinases exhibited a small, but systematic and statistically significant shift towards 17-AAG induced destabilization, with 18 kinases being significantly

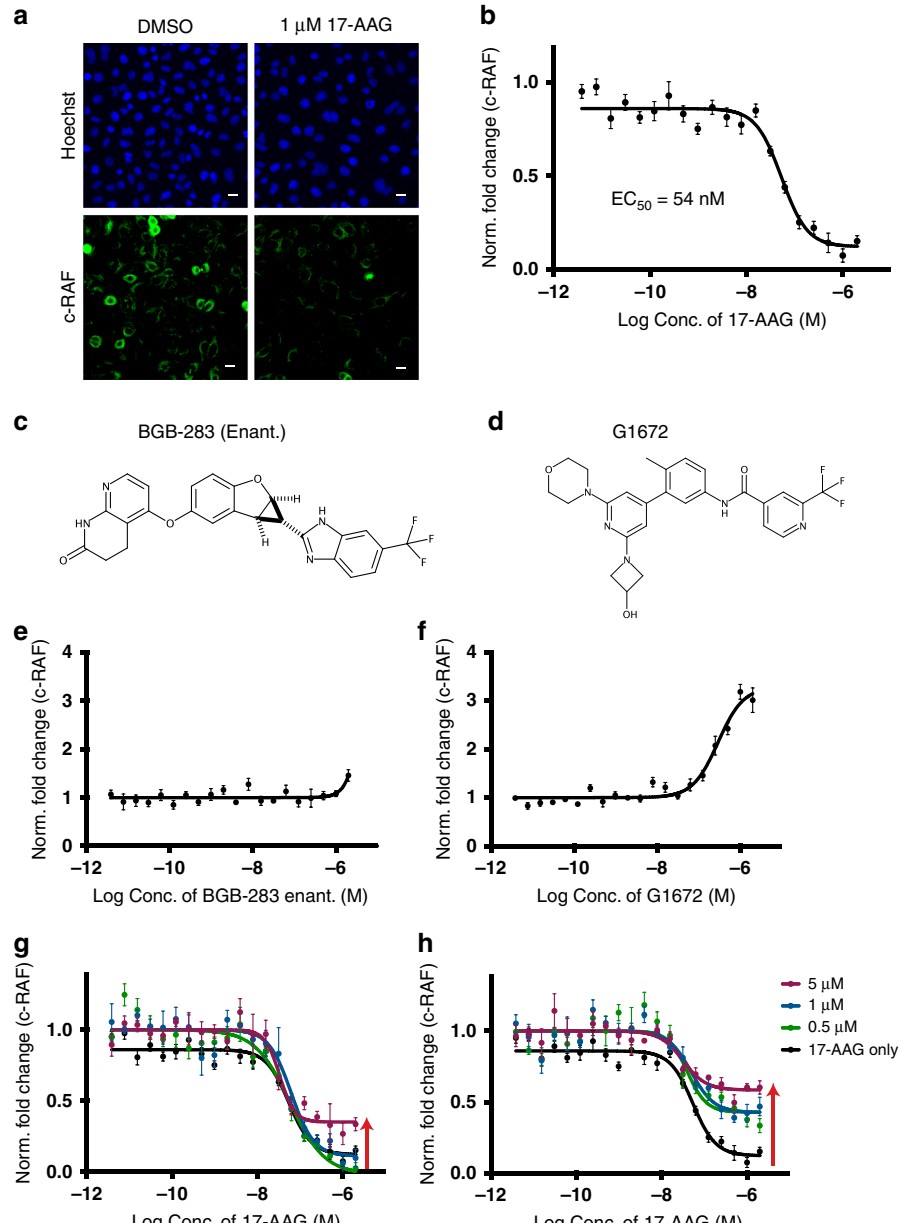

**Fig. 2 Quantitative fluorescence imaging measuring the effect of 17-AAG on c-RAF. a** Immunofluorescence images of c-RAF protein in A549 cells treated with DMSO or 17-AAG (1 μM) for 4 h. The upper panels show Hoechst 33342 staining of the DNA (blue) indicating the nuclear region, while the lower panels show the c-RAF immunofluorescence stain. Scale bar −20 μm. **b** The change in mean c-RAF protein per cell (A549 cells) was calculated for a range of concentrations of 17-AAG and expressed relative to the c-RAF level in DMSO treated samples and the no primary antibody control sample. **c** Chemical structures of c-RAF inhibitors: BGB-283 (enantiomer), **d** Novartis pan-RAF inhibitor. **e** Measurement of the relative c-RAF protein levels in A549 cells treated with a titration BGB-283 or **f** Novartis pan-RAF inhibitor. c-RAF protein levels are expressed as the fraction of c-RAF present in DMSO treated samples. The interactions between c-RAF and these inhibitors are examples of a small molecule causing a direct effect on protein abundance; c-RAF protein levels are elevated at high concentrations of the small molecule even in the absence of 17-AAG treatment. **g** Measurement of the relative c-RAF protein levels in A549 cells pre-treated with a range of concentrations (indicated by color) of BGB-283 or **h** Novartis pan-RAF inhibitor then treated with a titration of 17-AAG (x-axis) for 4 h. c-RAF levels are expressed as fold change in c-RAF levels relative to the DMSO treated sample. Error bars represent standard error of mean ($n = 3$). Source data are provided as a Source Data file.

down-regulated after 6 h and 27 kinases significantly down-regulated after 8 h (Fig. 4b and Supplementary Fig. 4A). Furthermore, other previously annotated HSP90 interactors[36–38] were also significantly down-regulated (Fig. 4c and Supplementary Fig. 4B).

A comparison of the biochemical inhibition profile of TAK-285 (across a panel of 235 protein kinases) suggests that approximately seven kinases are potential targets of TAK-285, with the most significant activity against EGFR, ERBB2, ERBB4,

and Lck (Supplementary Data 1). To determine which of these bind to TAK-285 in cells, we examined the proteins that were stabilized by TAK-285 in the context of HSP90 inhibition (i.e., proteins with a significant HIPStA response). The relative changes in protein abundance for samples pre-treated with TAK-285 prior to 17-AAG treatment, relative to samples that were treated with 17-AAG alone are depicted in Fig. 4d, e. The protein with the greatest HIPStA response to TAK-285 was HER2 (ERBB2). 17-AAG induced a significant reduction in

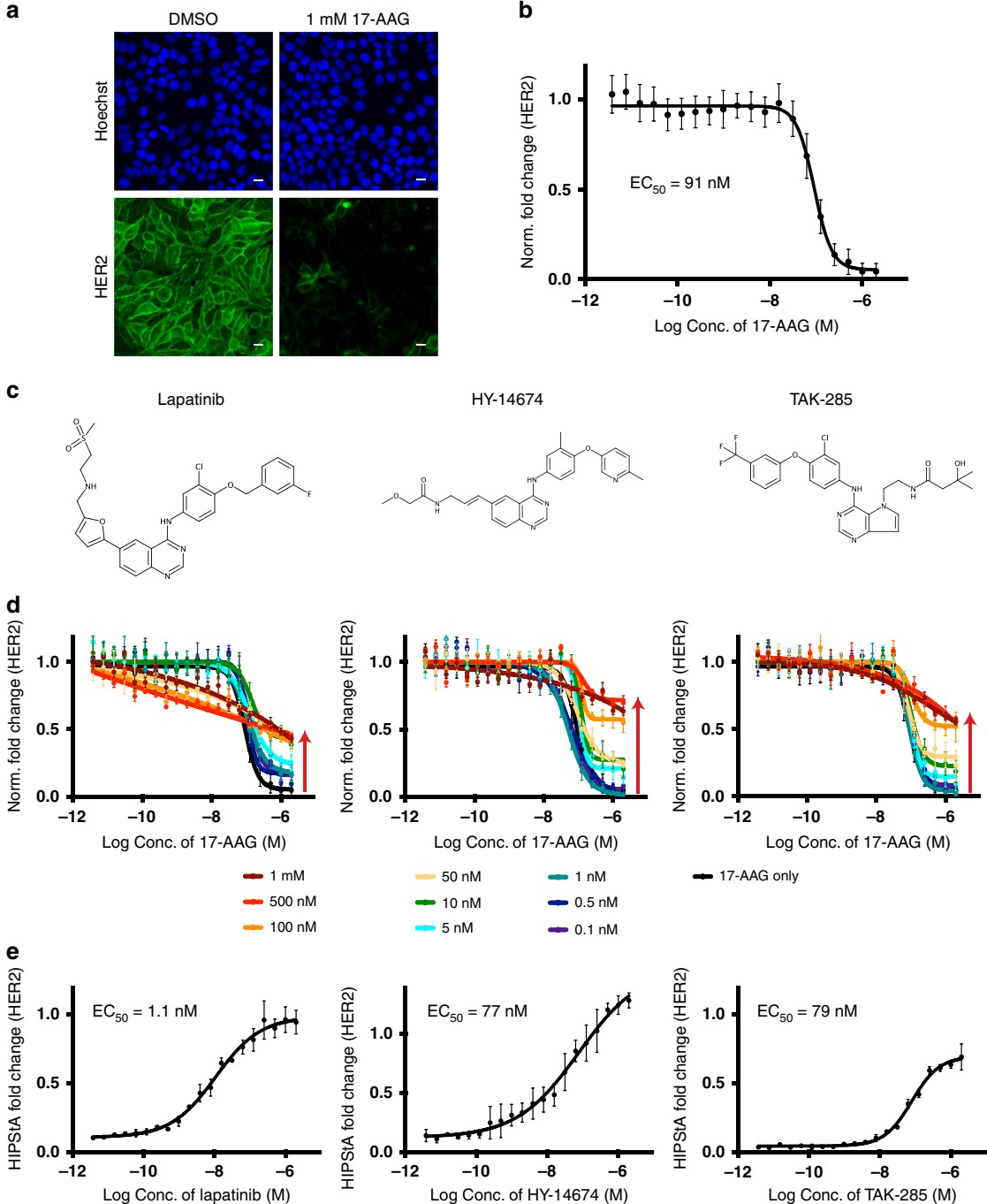

**Fig. 3 Quantitative fluorescence imaging measuring the effect of 17-AAG on HER2. a** Immunofluorescence images of HER2 protein in MCF-7 neoHER2 cells treated with DMSO or 17-AAG (1 μM) for 4 h. The upper panels show Hoechst 33342 staining of the DNA (blue) indicating the nuclear region, while the lower panels show the HER2 immunofluorescence stain. Scale bar −20 μm. **b** The change in mean HER2 protein per cell (MCF-7 neoHER2) was calculated for a range of concentrations of 17-AAG and expressed relative to the HER2 level in DMSO treated samples and the no primary antibody control sample. **c** Chemical structures of HER2 inhibitors: Lapatinib, HY-14674, and TAK-285. **d** Measurement of the relative HER2 protein levels in MCF-7 neoHER2 cells pre-treated with a range of concentrations of HER2 inhibitors (indicated by color) then treated with a titration of 17-AAG (x-axis) for 4 h. HER2 levels are expressed as normalized fold change in HER2 levels relative to the DMSO treated sample (Norm. Fold Change). **e** Measurement of the relative HER2 protein levels in MCF-7 neoHER2 cells pre-treated with a titration of each HER2 inhibitor then treated with 2 μM 17-AAG. HER2 protein levels are expressed as the fraction of HER2 present in the co-treatment condition relative to the corresponding concentration of inhibitor only (HIPStA Fold Change). The relative $EC_{50}$ values were calculated using a four parameter quadratic curve fit in PRISM. Error bars represent standard error of mean ($n = 3$). Source data are provided as a Source Data file.

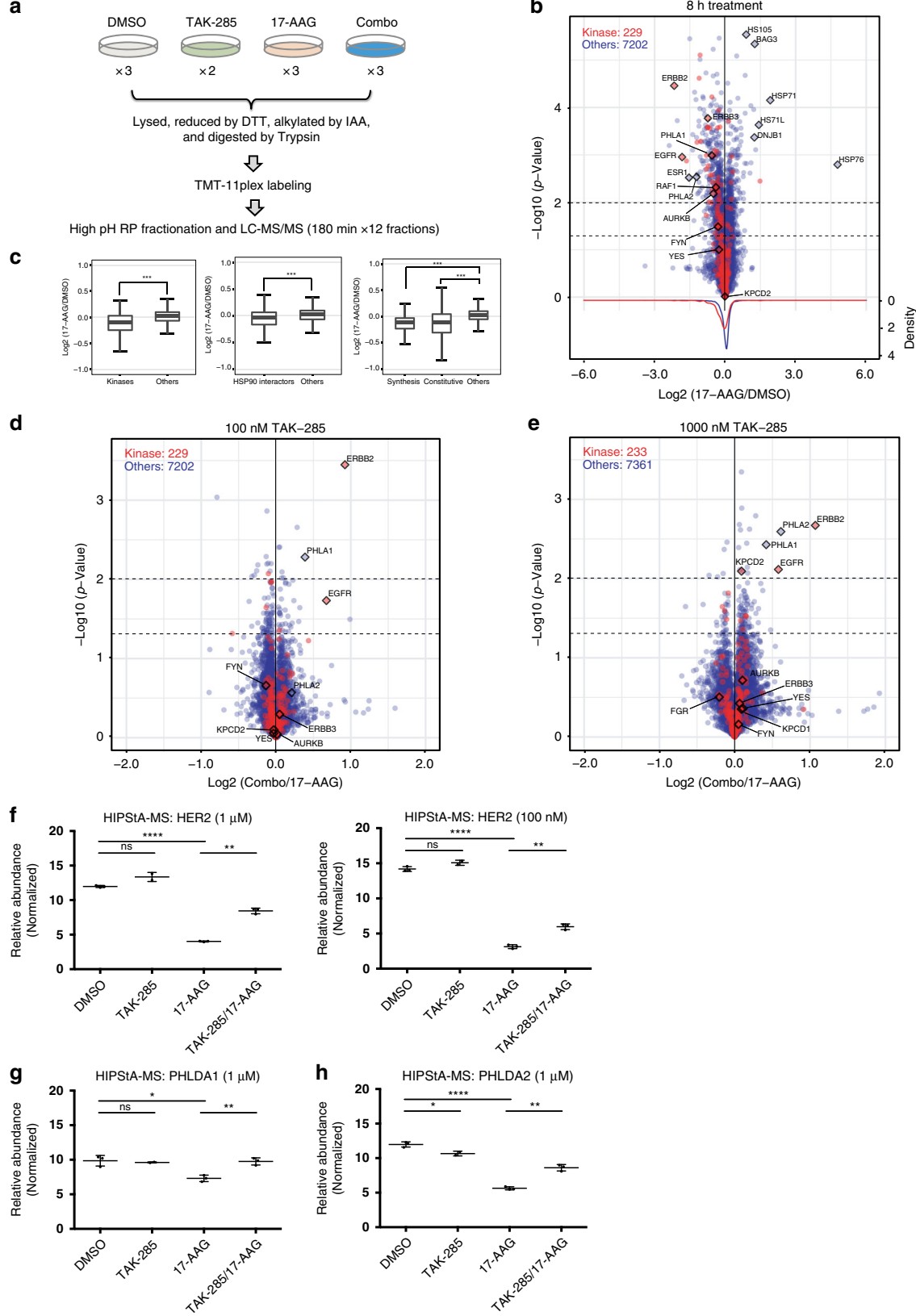

HER2 protein levels, which was partially protected against by pre-incubation with TAK-285 at 100 nM (its HIPStA EC$_{50}$ value) and 1 μM (Fig. 4d–f). The detection of EGFR (another HER family member) was ambiguous because the peptides used to detect it were also present in ERBB2. ERBB3, also in the HER family, was destabilized by 17-AAG, but this destabilization was not rescued with TAK-285 (Supplementary Fig. 5F), suggesting that TAK-285 does not bind ERBB3. Similarly, ERα and c-RAF, consistent with our previous data, were shown to be destabilized by 17-AAG, but this destabilization was not rescued with

**Fig. 4 Mass spectrometry-based proteomics identifies TAK-285 cellular protein targets. a** Schematic illustration of experimental design for global proteomics profiling. MCF-7 neoHER2 cells were treated with DMSO, TAK-285 (100 nM or 1 μM), 17-AAG (1 μM), or pretreated with TAK-285 (100 nM or 1 μM), followed by 17-AAG (1 μM). 17-AAG treatment time is 8 h (for the 100 nM set) and 6 h (for the 1 μM set). **b** Volcano plot depicting the fold-change in protein abundance in MCF-7 neoHER2 cells treated with 17-AAG (8 h) relative to DMSO samples. Each circle represents an individual protein with the color representing protein class (kinase: red; others: blue). X-axis: log2 transformed ratio of the protein levels in the 17-AAG treated samples and the DMSO treated samples. Y-axis: −log10 transformed p-value (dotted lines indicate $p = 0.05$ and $p = 0.01$; two-tailed student's $t$-test). Density curves for the two protein groups are shown at the bottom. **c** Distribution of ratio changes between 17-AAG (8 h) and DMSO in protein groups: by protein function, HSP90-interaction, or HSP90 dependency ($p < 0.00001$ (***)) by Welch two-sample t-test (box plots show minimum, first quartile, median, third quartile, and maximum). **d, e** Volcano plot showing the fold change in protein abundance in MCF-7 neoHER2 cells treated with a combination of 17-AAG and TAK-285, 100 nM **d** or 1 μM **e** relative to 17-AAG only samples. Annotated proteins (and alternative gene names) include: ERBB2 (HER2), PHLA1 (PHLDA1), PHLA2 (PHLDA2), KPCD2 (PRKD2) and EGFR. **f** Quantitation of ERBB2 (HER2) protein relative abundance in the four treatment groups. **g, h** Relative abundance of PHLDA1 **g** and PHLDA2 **h** proteins in the four treatment groups (1 μM TAK-285). Individual data points in F-H are calculated as percentage of sum total TMT signal for the protein (across all treatment groups), and shown as mean with standard deviation indicated. The statistical significance of the relative abundance of each protein in the different samples were determined using two-tailed student's $t$-tests for TAK-285 vs. DMSO, 17-AAG vs. DMSO, and TAK-285 + 17-AAG vs. 17-AAG ($n = 2$ for TAK-285 alone; $n = 3$ for all other conditions; $p < 0.05$ (*), $p < 0.01$ (**), $p < 0.001$ (***), $p < 0.0001$ (****). Source data are provided as a Source Data file.

TAK-285 (Supplementary Fig. 5A, B). Other kinases that are biochemical targets of TAK-285 (Supplementary Data 1), such as Yes, Fyn, and Src, did not demonstrate a significant HIPStA response (rescue by TAK-285). This could be due to the minimal sensitivity of their protein levels to 17-AAG treatment (Fig. 4b and Supplementary Fig. 6A-C); however an alternative target engagement method, nanoBRET, also failed to detect the binding of TAK-285 to the tyrosine kinase Yes in cells (Supplementary Fig. 7A).

The HIPStA method is applicable when the proteins of interest are destabilized by HSP90 inhibition. While a HIPStA global proteomics study using the non-selective kinase inhibitor staurosporine demonstrated that pre-treatment with staurosporine (1 μM) prior to 17-AAG treatment elevated the level of 349 proteins ($p < 0.05$ (two-tailed student $t$-test)), 47 of those proteins exhibited a significant reduction by 17-AAG ($p < 0.05$ (two-tailed student $t$-test); Supplementary Fig. 8B, 9 and Supplementary Data 2), with the most significant HIPStA response observed for 7 proteins that demonstrated both a greater than 15% reduction by 17-AAG ($p < 0.05$ (two-tailed student $t$-test)) and 15% stabilization by staurosporine ($p < 0.05$ (two-tailed student $t$-test)), including 5 kinases: ADRBK1 (ARBK1), ERBB2 (HER2), MASTL (GWL), STK32C and MAP3K20 (MLTK, ZAK); and 2 non-kinases (PAIP2 and RRM2 (RIR2)), (Supplementary Fig. 10 and Supplementary Data 2). Comparing our data with published CETSA based studies of staurosporine[39], we were limited by the 47% overlap between the 6286 fully annotated proteins detected in our global proteomics profiling study, and the fully annotated proteins detected in both overlapping studies used to define staurosporine binding (tested at 20 μM) by CETSA[39] (Supplementary Fig. 8A). Of the 7 proteins demonstrating a HIPStA response to staurosporine (1 μM), ADRBK1, MAP3K20 and MASTL were seen to be either stabilized or destabilized by staurosporine (20 μM) in the CETSA study, but only ADRBK1 met all selection criteria (Supplementary Data 2); ERBB2, PAIP2, RRM2 and STK32C were not detected in the overlapping CETSA mass spectrometry studies and hit designation was not defined[39]. Of the 60 proteins predicted by CETSA to bind staurosporine, 44 were also detected in our study with 3 (ADBRK1, GSK3A, and IRAK1) having >15% reduction caused by 17-AAG ($p < 0.05$ (two-tailed student $t$-test); Supplementary Fig. 8C; Supplementary Data 3) and therefore amenable to HIPStA. ADRBK1 was detected by HIPStA to be stabilized by staurosporine, but GSK3A and IRAK1 were not (Supplementary Data 3). Several other CETSA hits (CDK2, MAP2K1, MAP2K2, and PRKCE) were observed to be significantly stabilized by pretreatment with staurosporine prior to 17-AAG in the HIPStA study but by <15%; furthermore, PRKCD was observed to be

destabilized by staurosporine in the HIPStA study, consistent with the effect seen by CETSA (Supplementary Data 3 and Supplementary Fig. 10).

Interestingly, our global proteomics data for TAK-285 suggest that it also affects the stability of PHLDA1/PHLDA2, Pleckstrin homology-like domain family A proteins, which are implicated in the regulation of the Wnt3A β-catenin pathway and epithelial to mesenchymal transition[40,41] (Fig. 4g, h and Supplementary Fig. 5C, D). The independent identification of these two family members as proteins that are stabilized by TAK-285 contrasts with the third PHLDA family member, PHLDA3, which did not show a statistically significant decrease by 17-AAG, nor was it stabilized by TAK-285 (Supplementary Fig. 5E). The assignments of each of these proteins were with high confidence, due to all peptides being unique to their assigned proteins (Supplementary Fig. 11). Independent validation applying both HIPStA quantitative fluorescence imaging and CETSA methods, using two independent antibodies to detect PHLDA2, corroborate the observations from the mass spectrometry data indicating that TAK-285 modulates PHLDA2 protein stability; TAK-285 partially protects PHLDA2 from the effects of chaperone inhibition (Supplementary Fig. 12); however in the absence of HSP90 inhibition all three methods detect a destabilization of PHLDA2 by TAK-285 (Fig. 4h, Supplementary Fig. 12A–D, and Supplementary Fig. 13). While the functions of PHLDA proteins remain poorly understood, alignment of their protein sequences with that of HER2 indicates similarity to HER2's kinase domain (Supplementary Fig. 14). Our data demonstrate that the combination of HIPStA with a multiplexed global proteome profiling method identified, in addition to its known target HER2, the interaction of the small molecule TAK-285 with two previously unknown targets (PHLDA1 and PHLDA2).

## Discussion

Both mechanistic biological research and drug development benefit greatly from the ability to identify targets of a given small molecule and to quantitatively measure their binding affinities in the cellular context. In this study, we demonstrated that quantitative fluorescence imaging and quantitative mass spectrometry can be utilized with HIPStA to study small molecule interactions with endogenous targets in cells. HIPStA, which takes advantage of the change in a protein's stability upon ligand binding, utilizes pharmacological inhibition of the chaperone HSP90 to induce protein instability and avoids the use of heat, specialized probes, or protein tags, as used in other methods. We demonstrated proof-of-concept for measuring binding of small molecules to three clinically relevant targets of diverse subcellular localization:

ERα, c-RAF, and HER2. We successfully determined the HIPStA $EC_{50}$ for compounds targeting ERα and HER2, but measurements for c-RAF were confounded as the compounds alone elevated c-RAF levels, demonstrating the importance of monitoring the effects of drugs both with and without HSP90 inhibition in order to detect complex patterns of protein behavior. Nevertheless, in all three examples, we found that the primary effect of drugs that stabilize their protein targets is the elevation of the 17-AAG-induced degradation plateau. This allows the use of a single maximally effective concentration of 17-AAG, combined with a titration of the test compound to determine the compound's protein stabilization. Our results demonstrate that measuring changes in cellular protein abundance in HIPStA studies using quantitative fluorescence imaging is a highly effective method that enables measurements at a single-cell level, and simultaneously allows monitoring of cell health during the experiment.

We extended the concept of HIPStA to global proteome profiling to provide an orthogonal detection method of known drug targets, to identify the HSP90-dependent proteome (in the cells under study), and to detect potentially novel drug-protein interactions. The ability to detect novel drug interactions would help address the issue of off-target drug effects and guide drug development. We first investigated the HSP90-dependent proteome to determine proteins that could be compatible with HIPStA. While we found that many previously annotated HSP90 interactors were down-regulated upon 17-AAG treatment, we also observed proteins whose levels were consistently up-regulated by 17-AAG treatment (>25% increase, $p < 0.05$ (two-tailed student $t$-test)) (Fig. 4b, Supplementary Fig. 4A, C). Gene ontology enrichment identified the most significantly enriched molecular functions among these up-regulated proteins as "unfolded protein binding" and "chaperone binding" (GO:51082 and GO:51087), including known chaperone proteins HSP76 (HSPA6), HSP71 (HSPA1A), HS71L (HSPA1L), HS105 (HSPH1), HSPB8, DNAJB1, and BAG3, all of which suggest an adaptive response to HSP90 inhibition. This suggests that combining the inhibition of HSP90 with inhibition of HSP70 may be a fruitful extension of the HIPStA technique. Our comparison of 6 and 8 h 17-AAG treatment suggests that a longer incubation is beneficial for detecting proteome destabilization events and the protective effect of a drug on target proteins. However, longer treatment times risk the confounding effects of processes other than protein-homeostasis (such as transcriptional responses). The optimal concentration of a test compound will depend on its affinity for the target protein(s) and global proteomics mass spectrometry studies require sufficient concentrations to achieve near-complete target engagement. The detection of the interaction of TAK-285 with PHLDA2 by global proteomics mass spectrometry was achieved at 1 μM, but not 100 nM; this is consistent with estimates of the affinity of TAK-285 for PHLDA2 (120–480 nM Supplementary Fig. 12).

Our HIPStA global proteome study using the HER2 inhibitor TAK-285 revealed the expected target HER2. Another protein that initially appeared to show a significant HIPStA response in the global proteome profiling was EGFR, a protein in the same gene family as HER2 and a known target of TAK-285. However, the peptides that were used to identify EGFR are also present in ERBB2, and so the data do not provide incontrovertible evidence of EGFR identification (Supplementary Fig. 11). While we did not detect a significant HIPStA response for other potential targets of TAK-285 (Yes, Fyn and Src; Supplementary Data 1), some (e.g. Src) did not show significant sensitivity to 17–AAG treatment alone (Fig. 4b and Supplementary Fig. 6A-C), a prerequisite for the HIPStA method. However the application of an alternative target engagement technique, nanoBRET, also failed to detect binding of TAK-285 to the tyrosine kinase Yes, suggesting that, indeed it is not a target for TAK-285 in the cellular context

(Supplementary Fig. 7). The comparison of our HIPStA global proteomics profiling study of staurosporine with previously published CETSA data[39] suggests that the HIPStA method is complementary to the CETSA method; while both methods detected some common kinases engaged by staurosporine (ADBRK1 being the most significant), the HIPStA method detected some proteins not detected by CETSA and vice versa. Certainly some of these differences are due to the extent of overlap (or non-overlap) between protein sets detected in the various studies, while some differences may also be due to the difference in the concentrations of staurosporine tested (1 μM in HIPStA vs. 20 μM in CETSA). It is also very likely that some differences are due to whether or not different proteins are dependent on HSP90 chaperone function (HIPStA) or their susceptibility to thermal denaturation (CETSA). These characteristics will impact the ability of each technique to detect a drug-target interaction. Our HIPStA global proteomics study also revealed as significant TAK-285 hits, two previously unknown targets, PHLDA1 and PHLDA2. The modulation of the stability of PHLDA2 by TAK-285 was confirmed using two complementary techniques (HIPStA immunofluorescence and CETSA ELISA). It is interesting to note that although TAK-285 partially protects PHLDA2 from the effects of HSP90 inhibition, in the absence of chaperone inhibition, TAK-285 results in a partial destabilization of PHLDA2, illustrating the potentially complex nature of the effects of drug-target binding on protein homeostasis. This destabilization effect of drug-target binding is similar to that observed for staurosporine on PRKCD[39] (Supplementary Fig. 10N) and the effect of camptothecin on its target TOP1[12]. HIPStA quantitative fluorescence imaging data (Supplementary Fig. 12) suggest that there are two populations of PHLDA2; in the presence of active HSP90 chaperone, ~50% of PHLDA2 is destabilized by TAK-285, while in the absence of HSP90 chaperone function, ~50% of PHLDA2 is stabilized by TAK-285. While the reduction in PHLDA2 levels caused by TAK-285 is detected both in the mass spectrometry data and imaging data (Fig. 4h and Supplementary Fig. 12), the degree of its reduction is larger in the imaging data. We hypothesize that this effect may be caused by the different peptides that are detected in each method. The two peptides used to detect PHLDA2 in the mass spectrometry study are both in the N-terminus of the protein, while at least one of the antibodies used to detect PHLDA2 in the quantitative fluorescence imaging study recognizes an epitope in the C-terminus. Partial cleavage of the protein may result in a greater loss of signal by one method vs. the other. Alternatively, ubiquitination of PHLDA2 may differentially affect the ability of the two techniques to detect protein levels. Both the known PHLDA2 C-terminal epitope detected by one of the antibodies and one of the peptides used to detect PHLDA2 in the MS study contain lysines that could be sites for ubiquitination during protein destabilization and induced degradation. In the cell lysate CETSA study (Supplementary Fig. 13), only a destabilizing effect was observed, suggesting that the dilution of associating cofactors or accessory proteins may modulate the effect that TAK-285 has on PHLDA2 stability. PHLDA1 and PHLDA2 are evolutionarily conserved proline-histidine rich nuclear proteins from the PH-like superfamily, with PHLDA1 being a follicular stem-cell marker associated with tumor grade and aggressive behavior in follicular and sebaceous tumors[42–44], and PHLDA2 being an imprinted gene that is a prognostic marker in osteosarcoma and a modulator of the Wnt3A β-catenin pathway, critical for mesenchymal transition[40,41]. While our studies suggest that TAK-285 interacts with PHLDA proteins, it remains unclear what the pharmacological consequences may be.

In summary, HIPStA is a novel, high-throughput technique designed to detect drug-target engagement in cells (when the stability of a protein is modulated by chaperone function), which

can be successfully paired with quantitative fluorescence imaging and quantitative mass spectrometry. Furthermore, we believe that HIPStA could be applied beyond the HSP90-dependent proteome by pharmacological inhibition of alternative chaperones, such as HSP70[45]. HIPStA's ability to measure drug-target binding to endogenous targets in cells and to identify previously unknown targets of small molecules in an unbiased manner makes it a promising tool for applications in both research and drug development.

## Methods

**Cell culture and drug treatment**. The A549 lung carcinoma (ATCC® CCL-185), SK-BR-3 breast adenocarcinoma (ATCC® HTB-30) and AU565 breast adeno-carcinoma (ATCC® CRL-2351) cell lines were obtained from Genentech's cell line repository (gCell) and originated from ATCC. The cancer cell line MCF-7-neoHER2 was obtained from Genentech's cell line repository (gCell) and is an in vivo-selected line that expresses high levels of HER2 (3+) driven from an expression plasmid carrying a neomycin resistance gene and has the activating E545K PI3K mutation present in the parental MCF-7 cell line (ATCC® HTB-22). Cancer cell lines (MCF-7-neoHER2, A549, SK-BR-3 or AU565) were seeded on day 1 at a density of 10,000 cells per well in 384-well poly-lysine coated tissue culture plate (Greiner # T-3101-4), in 50 μL/well RPMI (phenol red free) cell culture medium containing 10% fetal bovine serum (FBS), and L-glutamine. In ERα experiments, charcoal stripped FBS was used. On Day 2, test compounds were serially diluted with DMSO, in a Labcyte Echo Qualified 384-well polypropylene plate (P-05525). DMSO was added to designated control wells. Compounds and controls were dispensed into cell culture plates using a Labcyte Echo acoustic dispenser (the final dispensed volume of each control and each compound was 50 nL), either in triplicate or quadruplicate. Cell plates were incubated with test compounds at 37 °C, for 30 min prior to the addition of 17-AAG, which was prepared either as a serial dilution or as a single concentration in a Labcyte Echo Qualified 384-well polypropylene plate (P-05525), and dispensed into the cell culture plate using a Labcyte Echo acoustic dispenser (the final dispensed volume of each control and each compound was 50 nL) and incubated for the specified time (typically 4 to 6 h). Following the addition of both test compound and 17-AAG, the final DMSO concentration was 0.2% v/v.

**Quantitative fluorescence imaging of proteins**. Measurements were taken from distinct samples. Fixation and permeabilization were carried out using a Biotek EL406 plate washer and dispenser as follows: cells were fixed by addition of 15 μL of 16% paraformaldehyde (Electron Microscopy Sciences #15710-S) directly into the 50 μL cell culture medium in each well using the peristaltic pump 5 μL cassette on a Biotek EL406 (final concentration of formaldehyde was 3.7% w/v). Samples were incubated for 30 min. Well contents were aspirated and 50 μL/well of Phosphate Buffered Saline (PBS) containing 0.5% w/v bovine serum albumin, 0.5% v/v Triton X-100 (Antibody Dilution Buffer) was added to each well. Samples were incubated for 30 min. Well contents were aspirated and washed 3 times with 100 μL/well of PBS. Immunofluorescence staining for the proteins of interest was carried out using a Biotek EL406 plate washer and dispenser as follows. Supernatant was aspirated from the wells and 25 μL/well of primary antibody (ERα antibody, Santa Cruz Biotech, sc-8002; HER2 antibody, abcam, ab134182; CRAF antibody, abcam, ab50858; PHLDA2 antibodies, Sigma Aldrich, SAB 2501914; ThermoFisher, PA5-76870) diluted 1:500 in Antibody Dilution Buffer was dispensed. Samples were incubated for 2 h at room temperature. Samples were washed 4x with 100 μL/well of PBS. 25 μL/well of secondary antibody solution (Alexafluor 488 conjugate anti-mouse IgG (Life Technologies #A1202) or Alexafluor 488 conjugate anti-rabbit IgG (Life Technologies #A1206 or Alexafluor 488 conjugate anti-goat Life Technologies # A-11055) diluted 1:1000 and Hoechst 33342 1 μg/mL diluted in Antibody Dilution Buffer) were dispensed into each well. Samples were incubated for 2 h at room temperature. Samples were washed 3x with 100 μL/well of PBS using a Biotek EL406. Quantitative fluorescence imaging of the proteins of interest was carried out using a CellInsight CX7 High-Content Screening (HCS) Platform (Thermo Fisher Scientific). Fluorescence images of the samples were acquired in 2 channels: Channel 1 - filter 386-23 BGRFRN_BGRFRN; Channel 2 - filter 485-20 BGRFRN_BGRFRN, using a fixed exposure time (based on DMSO control wells having an auto exposure peak target percentile of 25% target saturation) and 2 × 2 pixel binning. Channel 1 (DNA stain) was used to define the nuclear region (Circ). Protein average intensity measurements (AvgInt) were derived from the measurements of "Mean_CircAvgIntCh2", which is the average Alexafluor 488 fluorescence intensity (protein of interest) within the nuclear region or "Mean-RingAvgIntCh2" within the cytoplasmic or membrane region; intensities were measured on a per-cell basis and averaged over all the measured cells. Normalized Fold-change = $(F^{[17AAG, compound]}-F^{[max 17AAG]})/(F^{DMSO}-F^{[max 17AAG]})$; where $F^{[17AAG, compound]}$, $F^{[max 17AAG]}$, and $F^{DMSO}$ represent, respectively, the mean fluorescence average intensity values for protein levels in samples treated with corresponding concentrations of 17-AAG and specified test compound, or maximal 17-AAG concentration only, or DMSO only. HIPStA %fold-change = $(F^{[17AAG, compound]}-F^{[17AAG]})/(F^{[compound]} - F^{[17AAG]}) \times 100$; where $F^{[17AAG,}$

$^{compound]}$, $F^{[17AAG]}$, and F $^{[compound]}$ represent, respectively, the mean fluorescence average intensity values for protein levels in samples treated with the combination of a fixed concentration of 17-AAG and a specified concentration of test compound, or 17-AAG only or test compound only (at the specified concentration). Data analysis was carried out using Excel and Graphpad PRISM using a four parameter curve-fit (Y = Bottom + (Top-Bottom)/(1 + 10^((LogIC50-X)*Hill-Slope)) to define the $EC_{50}$.

**PHLDA2 CETSA ELISA**. Measurements were taken from distinct samples. MCF-7 neoHER2 cell lysates were prepared by resuspending cells ($10^8$ cells/mL) in PBS; then subjecting them to 3 cycles of freeze thaw and insoluble material was removed by centrifugation (16,000 × g, 30 min). Cell lysates were pre-incubated (1 h, 4 °C) with 10 μM TAK-285, or DMSO as control, and then dispensed into duplicate Eppendorf tubes (60 μL each). Samples were heated to defined temperatures (measured using a probe thermometer (Control Company, 4045DA) inserted into an Eppendorf tube) using a digital heat-block (VWR, 12621-092), then at each defined temperature, matched samples (+/− TAK-285) were transferred simultaneously to ice. Precipitated protein was removed by centrifugation (16,000 g, 15 min) and the PHLDA2 protein remaining in the supernatant was measured using a PHLDA2 sandwich enzyme linked immunosorbent assay (ELISA) using Corning Costar 96 well half area high binding ELISA plates (Corning #3690) coated with rabbit anti-PHLDA2 antibody (Life Technologies, PA5-76870 lot#UD2754884; 4 μg/ml in phosphate buffered saline (PBS) overnight at 4 °C), detection antibody goat anti-PHLDA2 (Sigma, SAB 2501914, lot# 6184P1; 5 μg/ml in PBS), and rabbit anti-goat IgG horseradish peroxide conjugate (Invitrogen, 61-1620, lot# TA257812; 0.75 μg/ml in PBS) detected using 1StepUltra TMB (Thermo, 34028 lot#UA2707432).

**Proteomic sample preparation**. Measurements were taken from distinct samples. MCF-7 neoHER2 cells were plated at 100% confluence in 10 cm² dishes and incubated overnight prior to addition of test compounds. Samples were prepared with either the DMSO control for baseline comparison, 17-AAG (1 μM), compound of interest (TAK-285 100 nM or 1 μM; or staurosporine 1 μM) or with a combination of the compound of interest pre-treatment followed by 17-AAG treatment (1 μM). A concentration of 1 μM 17-AAG was selected for the global proteomics profiling studies in order to achieve optimal protein destabilization at as low a concentration as possible (informed by evaluation of the immuno-fluorescence studies described in Figs. 1–3). Following an incubation period of 6 or 8 h, cells were snap-frozen and prepared for global proteome profiling. Cells were lysed in 8 M urea, 20 mM HEPES at pH 8.0 with sonication. Lysates were reduced with 4.5 mM dithiothreitol (DTT) at 65 °C for 25 min followed by alkylation with 11 mM iodoacetamide (IAA) at room temperature, in the dark for 20 min. The urea concentration was diluted to a final concentration of 2 M with 20 mM HEPES (pH 8.0). Proteins were digested with Lys-C (Wako) at an enzyme to substrate ratio of 1:50 and incubated at 37 °C for 3 h. Trypsin (Promega) was then added at a ratio of 1:100 and complete digestion was carried out overnight at 37 °C. Digests were acidified with trifluoroacetic acid, desalted using a C-18 Sep-pak (Waters), and dried. Each sample was resuspended with 100 μL of 100 mM HEPES pH 8.0 and labeled with a full vial of TMT10plex labeling reagent + TMT-131C labeling reagent (Thermo) for 1 h, according to the manufacturer's protocol. The reaction was quenched by adding 8 μL of 5% hydroxylamine, combined and desalted using a C-18 Sep-pak (Waters), and eluents were dried to completion.

**Offline high pH reversed phase fractionation**. High pH reversed phase liquid chromatography separation was applied to the previously labeled TMT tryptic peptides using an Agilent 1100 series HPLC system. Lyophilized peptides were resuspended in 0.1% TFA/water and loaded onto an Agilent Zorbax 300 Extend C-18 analytical column (2.1 × 150 mm and 3.5 μm particle size). Solvent A consisted of 25 mM Ammonium Formate (pH 9.7), while solvent B was 100% MeCN. Peptide fractions were collected in 45 s intervals for a total of 63 min with a linear gradient of 15–60% solvent B. In total 96 fractions were collected and every 25th fraction was combined to form a final set of 24 distinct groups. Samples were lyophilized, desalted in a STAGEtip, and injected for LC-MS/MS analysis.

**Mass spectrometric analysis of global proteins**. Data was collected on Orbitrap Fusion Tribrid or Orbitrap Fusion Lumos Tribrid Mass Spectrometers (Thermo Fisher Scientific). Peptides were loaded onto a New Objective PicoFrit Acquity® BEH130Å C18 column (1.7 μM, 100 μM × 250 mm) in Solvent A (98% water, 2% acetonitrile, 0.1% formic acid) via a Nanospray Flex Ion-Source (Thermo Scientific) at a voltage of 1.9 kV with a flow rate of 0.7 μL/min in Solvent A (98% water/2% acetonitrile/0.1% formic acid), separated at a flow rate of 0.5 μL/min with a linear gradient of 2 to 35% solvent B (98% acetonitrile/2% water/0.1% formic acid) over 158 min and sprayed into the mass spectrometer via a Nanospray Flex-Ion Source (Thermo Scientific) at a voltage of 1.9 kV. Full MS scans were collected in the orbitrap at 120,000 resolution, across a range from 350 to 1600 m/z, with an automatic gain control (AGC) target of $2 \times 10^5$ (Fusion) or $1 \times 10^6$ (Lumos), and a maximum injection time of 50 ms. MS² ions were selected in 0.7 Da (Fusion) or 0.5 Da (Lumos) isolation width with AGC of $5 \times 10^3$ (Fusion) or $2 \times 10^4$ (Lumos), and a maximum injection time of 100 ms using a top speed data dependent mode,

fragmented with CID energy of 35 and analyzed in ion trap. MS[3] spectra were acquired in the orbitrap by isolating 8 MS[2] ions in synchronized precursor selection (SPS) mode and fragmented via higher collision dissociation energy (HCD) of 55, AGC of $1 \times 10^5$ (Fusion) or $4 \times 10^5$ (Lumos), a maximum injection time of 150 ms, isolation width of 1.4 Da, and a resolution of 50,000 at 200 m/z.

**Mass spectrometric data identification and quantification**. MS/MS spectra were searched using Mascot (v.2.4.1) against UniProt DB (2015_04) with a taxonomy filter of '9606′, appended with common contaminating proteins and concatenated by all decoy sequences. Search parameters included trypsin cleavage with allowance of up to 2 missed cleavage events, a precursor ion tolerance of 50 ppm, and a fragment ion tolerance of 0.8 Da. Searches permitted variable modifications of methionine oxidation (+15.9949 Da), and static modifications of cysteine (+57.0215) for carbamidomethylation and TMT tags on lysine and peptide N termini (+229.1629). Peptide spectra matches (PSMs) were filtered with a false discovery rate (FDR) of 5% on the peptide level and subsequently at 2% on the protein level using linear discrimination. TMT reporter ions produced by the TMT tags were quantified with an in-house software package known as Mojave[46] by calculating the highest peak within 20 ppm of theoretical reporter mass windows and correcting for isotope purities. Quantified PSMs were filtered by total TMT reporter ion intensity greater than 50,000 and isolation specificity greater than 0.7, and then summarized to their matched proteins. Protein abundance ratios among different treatment groups were computed (as the percentage of sum total TMT signal for the protein across all treatment groups), in R (3.2.2) and p-values were calculated based on student t-test with two-tailed comparison assuming equal variance. For comparison of relative abundance values, values were normalized to the sum total of the protein detected across the different treatment sets for that protein, enabling the normalization of each individual measurement to an estimate of protein abundance derived from the largest possible sample size in order to provide a measure of relative changes in the level of each individual protein, while avoiding the dangers of normalizing to a relatively smaller sample set derived from just control (DMSO) samples. Gene Ontology enrichment analysis was performed using the 'hyperGTest' function in the R package 'GOstats' (version 2.48.0).

**Reporting summary**. Further information on research design is available in the Nature Research Reporting Summary linked to this article.

## Data availability

All mass spectrometry raw files have been deposited into the MassIVE database (http://massive.ucsd.edu/) and can be downloaded by the identifier MSV000084586 as well as in ProteomeXchange (http://www.proteomexchange.org/) with accession number PXD016301. The source data underlying Fig. 1b, d, e, 2b, e-h, 3b, d, e, 4b-g and Supplementary Figs. 1a, b, 2a, b, 3a-e, 4a-e, 5a-g, 6a-c, 7a, 8a-c, 9a-d, 10a-n, 12a-d, and 13a are provided as a Source Data File. All other data are available from the corresponding author upon reasonable request.

## Code availability

The Mojave software algorithm and functions used to process the TMT signal are available upon request.

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

## Acknowledgements

We thank Richard Neve, Klara Totpal, and Gail Phillips for their work confirming the identity and characteristics of the MCF-7 neoHER2 cell line and John Quinn, Chris Heise, and Margaret Scott for helpful discussions and their support for the research and this manuscript. This work was supported by Genentech Research and Early Discovery (gRED).

## Author contributions

K.F.C. and R.A.B. conceived the study. K.F.C. and R.A.B. performed quantitative immunofluorescence HIPStA and CETSA experiments and data analysis. T.P.M., C.M.R., D.S.K., and K.Y. performed quantitative mass spectrometry experiments and data analysis. K.F.C. and R.A.B. wrote the manuscript with contributions from all other authors. R.A.B. supervised all aspects of the work.

## Competing interests

Taylur P. Ma, Christopher M. Rose, Donald S. Kirkpatrick, Kebing Yu, and Robert A. Blake are all employees of Genentech (Roche USA) and own stock in Roche. Kelvin F. Cho was employed as an intern at Genentech (Roche USA) while contributing to this research.
