## [Peer Review File · Nature Communications]

Reviewers' comments:

Reviewer #1 (Remarks to the Author):

In their manuscript 'Chaperone mediated detection of drug target binding in cells' the authors use protein stability dependent on chaperone function to assess small molecule targets (called HIPStA). This is a further development of an approach developed by Taipale et al. which uses tagged proteins. The presented version overcomes these limitations and works in native cellular environment. In summary, proteins stabilized by HSP90 are destabilized with the use of an HSP90 inhibitor. Targets of an inhibitor of interest are protected from this destabilization effect and can thus be identified as such. Besides using quantitative fluorescence imaging for visualizing the reduction of known target protein levels, this assay can also be coupled with mass spectrometry to identify additional targets of an inhibitor.

The HIPStA assay thus adds to the arsenal of binding assays and, especially when coupled with mass spectrometry, can provide additional insight into target selectivity of small molecule inhibitors.

This offers a promising new approach for drug target identification however some concerns and suggestions need to be addressed by the authors.

1. As example, the authors chose HER2 inhibition by known HER2 inhibitors (Lapatinib, HY-14674 and TAK-285) in the MCF-7 breast cancer cell line. Using fluorescence readout, they also targeted two other classes of HSP90 clients, namely nuclear hormone receptors and cytoplasmic kinases. I suggest moving these results to a main figure because it shows broader application of the assay.

2. For the mass spectrometry experiment, the authors only chose the fairly selective, ATP competitive TAK-285 inhibitor. How does this assay coupled with MS readout perform for other inhibitors especially unselective kinase inhibitors like staurosporine or dasatinib? This might be a better example to investigate the overall performance of HIPStA. Does one recover most of the known targets?

3. Besides HER2, they identified two previously unknown, non-kinase off targets, PHLDA1 and PHLDA2. However, comparing the MS results to the selectivity profiling screen, Yes-kinase, which shows 86% inhibition is detected by MS but not identified as a potential target by the HIPStA assay. Does this imply the assay only recovers very strong binders? What are the detection limits of the assay and which affinities can be measured?

4. Does binding affinity correlate with degradation rate of HSP90?

5. How does this assay perform compared to other (kinase) binding and activity assays like CETSA, multiplexed inhibitor beads/kinobeads, KinomeScan?

6. Please update Supp Fig 5, the figure legend and volcano plot axes and titles do not match.

7. In Fig 2DE, also mark other TAK285 targets identified in the recombinant kinase panel assay.

8. Supp Fig 3. Panels E, F are duplicated.

9. I suggest moving spectrum in 2 F into supplement.

Reviewer #2 (Remarks to the Author):

The study by Cho et al. describes a new method for the identification of cellular targets of small molecule drugs, called HSP Inhibition Protein Stability Assay (Hipsta). This method is a further development of a recent approach that utilized the target stabilization by HSP90. Hipsta is following the same principle, but is furthermore based on the ability of drugs to stabilize their targets also in the presence of a HSP90 inhibitor, which by itself leads to destabilization of the target. The authors demonstrate the feasibility of this approach on several proteins, namely HER2, ERa and RAF1 using initially immunofluorescence detection. They also profile the HER2 inhibitor TAK-285 using a mass spectrometry-based approach.

This is an interesting study that showcases a promising new method towards drug target identification, which is primarily attractive due to the simplicity of the approach. Thus, it has the potential to be widely employed by many groups in different biological contexts. However, there are also some aspects that need further experimental elaboration and discussion to evaluate the general applicability of this method.

Major Critiques:

1. The observations made here would have to be shown also with other drugs, preferably those with a broad and well characterized target profile to deduce how widely applicable this method is. This will be informative particularly considering that only target proteins that are chaperoned by HSP90 will be captured by Hipsta, as the authors somewhat indicate correctly.
2. The cell line used in this study appears to be engineered to overexpress HER2. It would be important to demonstrate that Hipsta works equally well with endogenous protein using for instance an HER2 amplified cell line like SKBR3 or entirely different cells and targets. The authors show that Hipsta works also for Era and RAF1, but this is only shown by IF. It is not clear, if MS detection, which is necessary for unbiased identification of targets, would work equally well.
3. It is important to clearly state the strengths as well as the limitations of any new method. Thus it is important to state that Hipsta will only cover target proteins that are stabilized by HSP90 chaperoning although a drug may have multiple targets that do not depend on HSP90. In addition, the authors should discuss that with the current experimental design they won't be able to determine an IC50 from a mass spectrometry based experiment and would therefore require independent downstream validation of new targets. What is the IC50 value for EGFR as determined by Hipsta?

Minor Critiques:

1. Where in Figure 2 are ERBB4 and LCK? Are other known targets like PKD1/2 or YES observed by mass spec?
2. The concentration range queried for the BGB-283 compound is inadequate as it appears to be not potent enough for RAF1 and does barely protect RAF1 at 1 uM. Higher concentrations would need to be tested.
3. As most of the stabilization/destabilization effects observed are somewhat moderate, Figure 2 would be more informative if the x axes were rescaled with finer graduation.
4. Several supplementary figure (S4, S5C/D, S6A/B) are not discussed in the text although their inclusion does make sense and adds value to the manuscript.
5. Line 77: This refers to Figure 1E, not 2E.

6. Line 64: It would be more appropriate to talk of a half-life greater than 500 min instead of 511 as the experiment is ended at 500 min.
7. The double bond in Figure S2C needs to be redrawn.
8. The purpose of Figure S7 should be mentioned in the manuscript text.
9. Line 114: Please synchronize the nomenclature on PHLA1/2.
10. Line 114: This refers to Figure 2D-F, not 4D-F.
11. Line 210: TMT 10 plex or 11plex?
12. in Figure 2: Does the dashed horizontal line indicate a p value of 0.05?

Reviewer #3 (Remarks to the Author):

Peer review of manuscript for Nature Communications: NCOMMS-19-00133-T

Chaperone mediated detection of small molecule target binding in cells

Summary

This manuscript by Cho et al. describes a new approach for measuring cellular target engagement, which is an area of significant interest and with important applications in both drug discovery and chemical probe validation. The described methodology relies on the stabilizing effect (chaperone activity) of HSP90 for a subset of proteins. The authors demonstrate using three model systems that removal of this stabilizing effect, using an HSP90 small molecule inhibitor, can be exploited to improve the assay window for the established dynamic proteomics approach. Dynamic proteomics relies on the altered cellular turnover of proteins that is observed when ligands enter cells and bind to the intended target protein (see e.g. Cohen et al. in Science 1988 DOI: 10.1126/science.1160165 and abundant follow-up literature). While the original study tagged a significant portion of the proteome with a fluorescent protein, additional screen-compatible variants are available from e.g. DiscoverX using enzyme fragment complementation (InCell Hunter) to facilitate the readout of protein levels. Here the authors use immunofluorescence and mass spectrometry to allow probing of endogenous protein without any tags.

While dynamic proteomics is not applicable to all proteins, simple because the changes in total protein levels in response to ligand binding are too small, this manuscript describes an elegant way to open the assay window for HSP90 substrates. Despite this limitation (only applicable to a subpopulation of the proteome) the presented work can potentially be extended to more proteins by targeting other chaperones and other proteins that affect cellular protein turnover. I thus consider this work to be of general interest to this field and with obvious applications.

Recommendation: I feel the described approach represents a valuable addition to the literature and the work is clearly described in the manuscript and supporting figures. However, the manuscript as presented is incomplete and needs significant revisions as detailed below. The extent of the required revisions is such that I recommend rejection of the manuscript.

Specific feedback on content:

1. The manuscript is immature and lacks appropriate subheadings to facilitate reading. For example, the abstract is very short and lack details on the results or the potential future application of the methodology. Furthermore, the background description on alternative methodologies is limited to a few examples and in some cases incorrect. To exemplify: CETSA has clear limitations but the description of a complex mixture of concentrations and temperatures is not correct. There is ample literature showing how it can be readily applied for both screen purposes and for full concentration response profiling (see e.g. the recent publication from Shaw et al. in SLAS Discovery and last years publication from NCATS NIH on 17 model systems besides the cited literature). The concerns with CETSA are rather related to the need to heat cell samples, resulting in potential impacts on cell permeability and alterations of established equilibria. Other techniques for cellular target engagement, such as NanoBRET, FLIM, and dynamic proteomics (InCell Hunter), are left out. The lack of citations on the literature on dynamic proteomics is difficult to understand, I feel it should be mentioned at is relies on the same underlying concept, i.e. ligand induced alterations in cellular protein turnover. The authors should consider extending the background description and include these related approaches to put their work in perspective.

2. The authors claim that prior titration of a suitable 17-AAG concentration results in an assay with significantly improved throughput (page 3, line 74-75). It appears such titration is analogous the thermal aggregation curve that is established prior to HT CETSA experiments (to define a single temperature for profiling purposes), the titration of a fluorescent tracer concentration in NanoBRET experiments for use in competition experiments and a suitable endpoint time in dynamic proteomics (InCell Hunter) experiments. All these then allow characterization of full concentration responses so any throughput advantages should be more clearly described if the argument remains. The methodology is clearly complementary and may be responsive in cases where dynamic proteomics is not, but I am unable to understand the throughput argument.

3. A proteomics experiment is then used to examine to what extent the methodology can be applied for identification of target proteins in a cellular setting. However, this experiment is based on a single HSP90 inhibitor. The authors should consider validating their findings using additional

inhibitors or other means to suppress HSP90 activity. This is especially important as 17-AAG is also applied at a single concentration (1 μ M) for the proteomics experiments and used to support arguments that what is identified is the HSP90 client population (page 4, row 94). This claim needs to be supported by more rigorous comparisons with previous literature (the volcano plots in supplementary materials do not clearly demonstrate the correlation between herein achieved results and previous literature. Some support comes from the observation of kinase destabilization (broadly), but also this comparison needs to be on a more detailed level to understand correlations with prior data (or alternatively measured using an orthogonal approach).

4. Similarly, biochemical data highlight the interaction of TAK-285 with seven kinases, but the only example that is highlighted as a hit in the proteomics experiment is HER2, which coincides with what was used to first validate the approach. Although it is fully understandable that not all seven kinases are targeted in a cellular setting, the authors should make this comparison, clearly illustrate the data and allow the readers to form their own opinion.

5. Along the same lines the identification of PHLDA1/PHLDA2 as potential binders of TAK-285 in a cellular setting is interesting. However, the confirmation of this observation is only achieved by means of changing detection methodology, i.e. from MS to immunofluorescence. Given that this is a new approach, it would be useful to validate these findings using e.g. biochemical assays. Certainly, to achieve the standard of Nature Communications, several different approaches to validate the outcome can be expected. Possibilities besides biochemical assays on recombinant proteins include for example CETSA or resistance to proteolysis in cell lysates (DARTS).

6. The discussion is missing.

Point-by-point response to reviewer comments:

Reviewer #1

As example, the authors chose HER2 inhibition by known HER2 inhibitors (Lapatinib, HY-14674 and TAK-285) in the MCF-7 breast cancer cell line. Using fluorescence readout, they also targeted two other classes of HSP90 clients, namely nuclear hormone receptors and cytoplasmic kinases. I suggest moving these results to a main figure because it shows broader application of the assay.

We agree with the Reviewer and have moved the data on ER α and cRAF to the main figures to show generalizability to other HSP90 clients.

For the mass spectrometry experiment, the authors only chose the fairly selective, ATP competitive TAK-285 inhibitor. How does this assay coupled with MS readout perform for other inhibitors especially unselective kinase inhibitors like staurosporine or dasatinib? This might be a better example to investigate the overall performance of HIPStA. Does one recover most of the known targets?

To address this, we have performed an additional mass spectrometry experiment using staurosporine. From our results, we observe elevated protein levels of 39 kinases and observe the most significant HIPStA response in 4 kinases (MAP3K20 (ZAK), ERBB2, STK32C and MASTL (GWL)). For other proteins, we observe results that are consistent with some, but not all, observations from previous CETSA studies (Supp. Figure 8 in revised manuscript).

Besides HER2, they identified two previously unknown, non-kinase off targets, PHLDA1 and PHLDA2. However, comparing the MS results to the selectivity profiling screen, Yes-kinase, which shows 86% inhibition is detected by MS but not identified as a potential target by the HIPStA assay. Does this imply the assay only recovers very strong binders? What are the detection limits of the assay and which affinities can be measured?

The reviewer raises the question of whether the fact that we did not detect the interaction of TAK-285 with Yes kinase (a biochemical target of TAK-285) suggests that the technique will only detect very strong binders. To address this we applied the alternative target engagement method, nanoBRET, using nanoluciferase tagged Yes kinase and an active site tracer. Our data indicate that the nanoBRET method was also unable to detect the binding of TAK-285 to Yes kinase, in contrast to our positive control dasatinib (Supp. Fig. 7 in revised manuscript). These results support the conclusions of the HIPStA study, that TAK-285 does not interact with Yes kinase in the cellular context. However we should also comment that, from our MS results, we observed that the levels of Yes kinase did not change drastically with 17-AAG treatment alone (Fig. 4B and Supp. Fig. 6A-C in revised manuscript). This would result in a diminished possible dynamic range for rescue by TAK-285, compared to HER2.

Does binding affinity correlate with degradation rate of HSP90?

While we have not investigated degradation of HSP90 directly, an increased degradation rate of HSP90 would likely have the same effect as increasing 17-AAG, since both result in a decrease of functional HSP90. In our case, since we are using the same cells within each experiment and assume the same rate of protein degradation, we attribute the change in our measurements to 17-AAG's inhibition of HSP90's chaperone function.

How does this assay perform compared to other (kinase) binding and activity assays like CETSA, multiplexed inhibitor beads/kinobeads, KinomeScan?

To address this question of how HIPStA compares to other binding techniques, we carried out CETSA studies of the interaction of TAK-285 with PHLDA2, which was initially identified as a potential TAK-285 interacting protein in the HIPStA studies. Our data corroborate the observations in the HIPStA study, that TAK-285 modulates PHLDA2 stability (Supp. Figure 11 in revised manuscript). In an additional staurosporine HIPStA global proteomics profiling experiment (Supp. Figure 8 in revised manuscript), we identify several proteins that were also identified in a previously published CETSA study (but not all), as well as several additional potential targets of staurosporine. We also confirm that TAK-285 does not bind to YES kinase (as suggested by HIPStA) using a nanoBRET based binding assay.

Please update Supp Fig 5, the figure legend and volcano plot axes and titles do not match.

We have fixed this and have changed the axes and titles to be consistent with the figure legends, thank you.

In Fig 2DE, also mark other TAK285 targets identified in the recombinant kinase panel assay.

We have updated Fig. 2BDE (Fig. 4 in revised manuscript) to include Lck, Yes, Fyn, Fgr, PKD1, PKD2, and AurKB (given they were detected in the respective experiment).

Supp Fig 3. Panels E, F are duplicated.

The Panel E and F of Supp. Fig. 3 (Fig. 2 in revised manuscript) show c-RAF levels when cells are treated with either the BGB283 (enantiomer) compound (E) or the Novartis pan-RAF inhibitor (F) and represent two different sets of experiments/results.

I suggest moving spectrum in 2 F into supplement.

We have moved the spectrum in Fig. 2F (Fig. 4F in revised manuscript) to the supplement, thank you for the suggestion.

Reviewer #2

The observations made here would have to be shown also with other drugs, preferably those with a broad and well characterized target profile to deduce how widely applicable this method is. This will be informative particularly considering that only target proteins that are chaperoned by HSP90 will be captured by Hipsta, as the authors somewhat indicate correctly.

In a similar response to the previous reviewer's concern, we have performed an global proteomics profiling experiment using staurosporine. We observe elevated protein levels of 39 kinases and observe the most significant HIPStA response in 4 kinases (MAP3K20 (ZAK), ERBB2, STK32C and MASTL (GWL)). For other proteins, we observe results that are consistent with some, but not all, observations from previous CETSA studies (Supp. Figure 8 in revised manuscript) (Supp. Figure 8 in revised manuscript).

The cell line used in this study appears to be engineered to overexpress HER2. It would be important to demonstrate that Hipsta works equally well with endogenous protein using for instance an HER2 amplified cell line like SKBR3 or entirely different cells and targets. The authors show that Hipsta works also for Era and RAF1, but this is only shown by IF. It is not clear, if MS detection, which is necessary for unbiased identification of targets, would work equally well.

We have repeated our assay using TAK-285 in two other cell lines SK-BR-3, as suggested by the reviewer, and AU565. We observe that TAK-285, but not dacomitinib, protects against 17-AAG induced degradation of HER2, consistent with our observations in the MCF7 neoHER2 line (Supp. Figure 3 in revised manuscript). Furthermore, our new mass spectrometry data using staurosporine also addresses this point as we detected HIPStA responses for a range of proteins using the non-selective kinase inhibitor. This data suggests that HIPStA is sensitive enough to identify endogenous targets.

It is important to clearly state the strengths as well as the limitations of any new method. Thus it is important to state that Hipsta will only cover target proteins that are stabilized by HSP90 chaperoning although a drug may have multiple targets that do not depend on HSP90. In addition, the authors should discuss that with the current experimental design they won't be able to determine an IC50 from a mass spectrometry based experiment and would therefore require independent downstream validation of new targets. What is the IC50 value for EGFR as determined by Hipsta?

We have included in our discussion that HIPStA in its current state is limited to HSP90 clients, but that it could also potentially be adapted to surveying clients of other chaperones. We present HIPStA paired with mass spectrometry as a strategy for identifying targets of a drug in an unbiased manner. However, we also show using multiple approaches, as in the case of PHLDA2, that additional biochemical characterization is required for validation of identified targets.

Where in Figure 2 are ERBB4 and LCK? Are other known targets like PKD1/2 or YES observed by mass spec?

We have updated Fig. 2BDE (Fig. 4 in revised manuscript) to include YES, FYN, FGR, PRKD1 (KPCD1), PRKD2 (KPCD2), and AURKB (given they were detected in the respective experiment). While these proteins were detected by mass spectrometry, we did not observe a decrease by 17-AAG that was rescued by TAK-285. ERBB4 and LCK were not detected in these experiments.

The concentration range queried for the BGB-283 compound is inadequate as it appears to be not potent enough for RAF1 and does barely protect RAF1 at 1 μ M. Higher concentrations would need to be tested.

For BGB-283 (Enant.), we tested concentrations up to 5 μ M. We observe a protection effect as shown by the elevation of the curve plateau (Figure 2G in revised manuscript). We do note that at high concentrations of BGB-283 (Enant.), we observe an effect on increased c-RAF levels (Figure 2E in revised manuscript), which can confound HIPStA measurements.

As most of the stabilization/destabilization effects observed are somewhat moderate, Figure 2 would be more informative if the x axes were rescaled with finer graduation.

To maintain figure clarity, we did not add finer gradation to the x-axis. However, if a reader wishes to more carefully analyze the data, the entries with significant responses for the mass spectrometry experiment using TAK-285 can be found in Supplementary Table 2.

Several supplementary figure (S4, S5C/D, S6A/B) are not discussed in the text although their inclusion does make sense and adds value to the manuscript.

We have included discussion of these supplementary figures in the main text to support our claims, thank you.

Line 77: This refers to Figure 1E, not 2E.

We have corrected this figure reference.

Line 64: It would be more appropriate to talk of a half-life greater than 500 min instead of 511 as the experiment is ended at 500 min.

We have changed the half-life to >500 minutes, as suggested, thank you.

The double bond in Figure S2C needs to be redrawn.

We have included a new structure for Endoxifen in Figure S2C (Fig. 1C in revised manuscript).

The purpose of Figure S7 should be mentioned in the manuscript text.

We have included reference to this figure (Supp. Fig. 9 in revised manuscript) to show confidence in our assignments of each of these proteins, due to unique peptides between each.

Line 114: Please synchronize the nomenclature on PHLA1/2.

We have changed the manuscript to refer to the proteins as PHLDA1/2 consistently.

Line 114: This refers to Figure 2D-F, not 4D-F.

We have corrected this figure reference, thank you.

Line 210: TMT 10 plex or 11plex?

Our samples were prepared with the TMT10plex labeling reagent + TMT11 reagent set from Thermo. We have updated the methods to clarify this.

in Figure 2: Does the dashed horizontal line indicate a p value of 0.05?

In Figure 2 (Figure 4 of revised manuscript), the y-axis is the $-\log(p\text{-value})$, so the bottom dashed line indicates a p-value of 0.05 and the top dashed line indicates a p-value of 0.01.

Reviewer #3

The manuscript is immature and lacks appropriate subheadings to facilitate reading. For example, the abstract is very short and lack details on the results or the potential future application of the methodology. Furthermore, the background description on alternative methodologies is limited to a few examples and in some cases incorrect. To exemplify: CETSA has clear limitations but the description of a complex mixture of concentrations and temperatures is not correct. There is ample literature showing how it can be readily applied for both screen purposes and for full concentration response profiling (see e.g. the recent publication from Shaw et al. in SLAS Discovery and last years publication from NCATS NIH on 17 model systems besides the cited literature). The concerns with CETSA are rather related to the need to heat cell samples, resulting in potential impacts on cell permeability and alterations of established equilibria. Other techniques for cellular target engagement, such as NanoBRET, FLIM, and dynamic proteomics (InCell Hunter), are left out. The lack of citations on the literature on dynamic proteomics is difficult to understand, I feel it should be mentioned at is relies on the same underlying concept, i.e. ligand induced alterations in cellular protein turnover. The authors should consider extending the background description and include these related approaches to put their work in perspective.

We have expanded our Abstract, included appropriate subheadings in our Results, and have included a Discussion section. We have also extended our Background section to include further adaptations of CETSA as well as other techniques that the Reviewer has mentioned. We include

the overall limitations of these other approaches and how HIPStA overcomes some of these issues. Thank you for the feedback.

The authors claim that prior titration of a suitable 17-AAG concentration results in an assay with significantly improved throughput (page 3, line 74-75). It appears such titration is analogous the thermal aggregation curve that is established prior to HT CETSA experiments (to define a single temperature for profiling purposes), the titration of a fluorescent tracer concentration in NanoBRET experiments for use in competition experiments and a suitable endpoint time in dynamic proteomics (InCell Hunter) experiments. All these then allow characterization of full concentration responses so any throughput advantages should be more clearly described if the argument remains. The methodology is clearly complementary and may be responsive in cases where dynamic proteomics is not, but I am unable to understand the throughput argument.

We agree that alternative methods can also be adapted to have high throughput. We have adjusted our text to include this. While HIPStA offers the advantage of being a high-throughput technique, we no longer claim that this is unique to HIPStA, and we agree that HIPStA is complementary to the other techniques. We have included this in our Background and Discussion sections.

A proteomics experiment is then used to examine to what extent the methodology can be applied for identification of target proteins in a cellular setting. However, this experiment is based on a single HSP90 inhibitor. The authors should consider validating their findings using additional inhibitors or other means to suppress HSP90 activity. This is especially important as 17-AAG is also applied at a single concentration (1 μ M) for the proteomics experiments and used to support arguments that what is identified is the HSP90 client population (page 4, row 94). This claim needs to be supported by more rigorous comparisons with previous literature (the volcano plots in supplementary materials do not clearly demonstrate the correlation between herein achieved results and previous literature. Some support comes from the observation of kinase destabilization (broadly), but also this comparison needs to be on a more detailed level to understand correlations with prior data (or alternatively measured using an orthogonal approach).

We based our HIPStA assay development on 17-AAG since it is widely available to researchers, and the geldanamycin analogs, such as 17-AAG, are probably the best characterized of the HSP90 inhibitors, inhibiting all four mammalian isoforms of HSP90. While there are some off-target effects known for geldanamycin analogs, they are well characterized (unlike many other HSP90 inhibitors) and typically become apparent at higher concentrations (>10 μ M) than used in this study (Neckers et al Cell Stress and Chaperones (2018) 23:467–482). However, to address the question of whether another HSP90 inhibitor could be used instead of 17-AAG, we carried out a HIPStA study of HER2 stabilization by TAK-285 in SK-BR-3 cells, using the tropanone-based HSP90 inhibitor XL888; the results are essentially the same as those for 17-AAG (Supp. Fig. 3 in revised manuscript). Also, in a similar response to the previous reviewers' concerns, we have performed an additional mass spectrometry experiment using the non-selective inhibitor staurosporine. We observe elevated protein levels of 39 kinases and observe the most significant HIPStA response in 4 kinases (MAP3K20 (ZAK), ERBB2, STK32C and MASTL (GWL)). For

other proteins, we observe results that are consistent with some, but not all, observations from previous CETSA studies (Supp. Figure 8 in revised manuscript).

Similarly, biochemical data highlight the interaction of TAK-285 with seven kinases, but the only example that is highlighted as a hit in the proteomics experiment is HER2, which coincides with what was used to first validate the approach. Although it is fully understandable that not all seven kinases are targeted in a cellular setting, the authors should make this comparison, clearly illustrate the data and allow the readers to form their own opinion.

As similarly requested by previous reviewers, we have updated Fig. 2BDE (Fig. 4 in revised manuscript) to include Yes, Fyn, Fgr, PRKD1 (KPCD1), PRKD2 (KPCD2), and AURKB (given they were detected in the respective experiment).

Along the same lines the identification of PHLDA1/PHLDA2 as potential binders of TAK-285 in a cellular setting is interesting. However, the confirmation of this observation is only achieved by means of changing detection methodology, i.e. from MS to immunofluorescence. Given that this is a new approach, it would be useful to validate these findings using e.g. biochemical assays. Certainly, to achieve the standard of Nature Communications, several different approaches to validate the outcome can be expected. Possibilities besides biochemical assays on recombinant proteins include for example CETSA or resistance to proteolysis in cell lysates (DARTS).

We have further validated our findings from the TAK-285 study using CETSA to assess the effect of TAK-285 on the thermal stability of PHLDA2 in cell lysates (Supp. Fig. 11 in revised manuscript). Consistent with our mass spectrometry results and immunofluorescence results, the CETSA data indicate that TAK-285 modulates PHLDA2 stability, although it is interesting to note that in the absence of HSP90 inhibition, TAK-285 destabilizes PHLDA2 in both CETSA and HIPStA studies.

The discussion is missing.

We have expanded our manuscript considerably and it now includes a discussion section, with key points that reviewers have addressed above, thank you.

Reviewers' comments:

Reviewer #1 (Remarks to the Author):

The revised manuscript successfully addresses most of the concerns raised in the previous round of review.

As one could foresee the introduce method (HIPStA) will be an addition to the toolkit used in chemical biology to identify targets of small molecules. I therefore support publication in Nature Communications.

To act as a guide for others, the current manuscript still lacks some discussion on method relevant parameters. The authors should mention details on dosage and time point selection. Is there a reason for different amounts of 17-AAG used in the immunofluorescence vs MS experiments? Along the same line, authors should address the combination of incubation time and drug concentrations used. Lines 152 to 160 read as if the higher number of kinases observed is related to the longer incubation period, whereas with 10x more drug targets do separate better from non-binders.

Reviewer #2 (Remarks to the Author):

The authors have addressed most of my previous critiques adequately, but not all. The most important remaining concern pertains to the degree of proteome coverage and the general utility of this approach, which suggests that this is an attractive method for select targets, particularly for in cell validation, but that it has only limited utility for identification of unknown drug targets or for comprehensively describing drug-target profiles.

1. Supp. Figure 8B: This is confusing: Is this staurosporine-treated as indicated by label above graph and figure legend? The data rather resembles what one would expect from 17-AAG treatment only.

2. Supp. Figure 8A: Why are there so few targets of staurosporine identified? In the Taipale et al. Nat Biotechnol 2013 paper, using a similar chaperoning approach, many more staurosporine targets are identified, including EGFR, which should be amenable to the authors' approach according to Figure 4. Several other targets are ubiquitously expressed and should be also detected (e.g. ABL1, AURKA+B, AMPK2,...). This suggests that the authors' approach described here provides only limited coverage of the proteome. No MS-based data is provided for ERα and RAF1, which could address this issue.

3. Supp. Figure 8A/B: Which cell line has been used?

4. In the rebuttal letter please indicate the specific changes made in the manuscript.
5. Line 264: The sentence is referring to Supp. Figure 9, not 7.

Reviewer #3 (Remarks to the Author):

As stated last time the manuscript is relevant to the field and presents interesting data. The manuscript has improved significantly since the last submission and is now well structured and complete. There are however a need for minor revisions to further improve the manuscript. These are outlined in the accompanying PDF file.

I recommend publication given considerations of these improvements.

NCOMMS-19-00133B

Chaperone mediated detection of small molecule target binding in cells
Cho et al.

Point-by-point response to reviewer comments:

Reviewer #1

To act as a guide for others, the current manuscript still lacks some discussion on method relevant parameters. The authors should mention details on dosage and time point selection. Is there a reason for different amounts of 17-AAG used in the immunofluorescence vs MS experiments?

We had initially performed our immunofluorescence studies for the fixed 17-AAG concentration experiments using 2 μ M 17-AAG. When progressing to our mass spec experiments, we noticed that even at concentrations of 1 μ M 17-AAG (or lower), such as in Figure 3D, fold change values were reaching a plateau. We selected a concentration of 1 μ M 17-AAG for the global proteomics profiling studies in order to achieve optimal protein destabilization at as low a concentration as possible (informed by evaluation of the immunofluorescence studies described in Figures 1-3); the text in the Figure 4 has been changed to clarify this.

Along the same line, authors should address the combination of incubation time and drug concentrations used. Lines 152 to 160 read as if the higher number of kinases observed is related to the longer incubation period, whereas with 10x more drug targets do separate better from non-binders.

We have added a discussion on the advantages and disadvantages of longer and shorter incubation times and the required concentration of test compounds to detect interactions with target proteins (lines 278 – 286) of the discussion section.

Reviewer #2

Supp. Figure 8B: This is confusing: Is this staurosporine-treated as indicated by label above graph and figure legend? The data rather resembles what one would expect from 17-AAG treatment only.

Agreed. This figure was labeled in a confusing manner. We have changed the title label on the graph in Supp. Figure 8B (now Supp. Figure 10B) to read “17-AAG 6 hours treatment”.

Supp. Figure 8A: Why are there so few targets of staurosporine identified? In the Taipale et al. Nat Biotechnol 2013 paper, using a similar chaperoning approach, many more staurosporine targets are identified, including EGFR, which should be amenable to the authors' approach according to Figure 4. Several other targets are ubiquitously expressed and should be also detected (e.g. ABL1, AURKA+B, AMPK2,...). This suggests that the authors' approach described here provides only limited coverage of the proteome. No MS-based data is provided for ERα and RAF1, which could address this issue.

We agree that there is a difference in the number of protein species detected between our approach and that of Taipale et al. *Nature Biotechnology*, 2013. However, the methods differ fundamentally in ways that would explain this difference. In Taipale et al., HEK293T cells stably expressing HSP90 fused to luciferase were transiently transfected with the kinase being assayed. The kinase was then immunoprecipitated and subsequent luciferase and ELISA assays were performed. In this setting, the kinase is tagged and overexpressed above endogenous levels (as mentioned in the Discussion of Taipale et al. *Nature Biotechnology*, 2013). Meanwhile, a key advantage of HIPStA is that our assay detects changes in endogenous untagged protein levels and does not rely on transfection of assay targets. Furthermore, Taipale et al. have enrichment and signal amplification from immunoprecipitation and ELISA that could explain higher sensitivity compared to the global proteomics profiling approach (whole cell lysates) in our study. In regards to staurosporine concentrations, Taipale et al. used 5 μM staurosporine in their studies, while we used 1 μM staurosporine, which is another possible source of difference between our results. We have also included a much more extensive comparison of our HIPStA global proteomics profiling study of staurosporine with the published CETSA study of staurosporine (Savitsky *et al*) with additional Supplementary Figures 8 and 9 and Supplementary Tables 3 and 4; and description of the results on lines 203-220 and discussion on lines 300-309. Again there were significant differences in the concentration of staurosporine used (20 μM in Savitsky *et al* vs 1 μM in our studies).

Specifically regarding the detection of staurosporine binding to EGFR, our initial assessment suggested that EGFR was a borderline hit in the HIPStA study where 17-AAG was significantly destabilizing and staurosporine elevated the protein levels but the effect did not quite reach significance ($p=0.08$); however on closer inspection three of four EGFR peptides were shared with ERBB2, and one EGFR unique peptide did not demonstrate quantitative changes among the four conditions. The apparent change in EGFR protein was driven by the peptides that could not be distinguished from ERBB2. We therefore did not include EGFR as a potential hit in our staurosporine HIPStA MS study. It should also be noted that the CETSA study of Savitsky et al. did not detect EGFR as a staurosporine interacting protein.

Regarding the question of why we have not discussed “several other targets (that) are ubiquitously expressed and should be also detected (e.g. ABL1, AURKA+B, AMPK2,...)” and whether we “provide only limited coverage of the proteome” – We hope the more expansive discussion of the overlap between our data and published CETSA data (outlined above and illustrated in additional

figures and tables) help address these concerns. To address more specifically the proteins mentioned by the reviewer: The new Supplementary Table 4 contains data for AURKA, which was detected in our study and showed a 23% reduction by staurosporine, but fell just short of statistical significance ($p=0.059$). ABL1 was also detected in our study but did not show either a significant reduction by 17-AAG nor a stabilization by staurosporine, but neither was it identified as a staurosporine interacting protein in the comparator CETSA study by Savitsky et al. (despite being detected). While AMPK2 (PRKAA2) was not detected in our study, the family member AMPK1 (PRKAA1) was, but it did not show stabilization by staurosporine (again corroborating the data in the comparator CETSA study by Savitsky et al.).

Supp. Figure 8A/B: Which cell line has been used?

We have modified the figure legend, which is now Supplementary Figure 10, to include the cell line information. Thank you for pointing this out.

Line 264: The sentence is referring to Supp. Figure 9, not 7.

This error has now been fixed in the main text, thank you. But please note that figure numbers have changed after new Supplementary Figures 8 and 9 were inserted

Reviewer #3

As stated last time the manuscript is relevant to the field and presents interesting data. The manuscript has improved significantly since the last submission and is now well structured and complete. There are however a need for minor revisions to further improve the manuscript. These are outlined in the accompanying PDF file.

I recommend publication given considerations of these improvements.

The content of the accompanying PDF file is pasted below. Our original comments and the summary statement from the reviewers are retained in black (non-italicized); the reviewers' comments are retained in *italic* and in **blue text**. Our new responses are highlighted in **green**.

Peer review of revised manuscript for Nature Communications: NCOMMS-19-00133-T - Chaperone mediated detection of small molecule target binding in cells; with author responses

The readability of the updated manuscript by Cho *et al.* has improved significantly and now includes a better structuring and a thorough discussion of the results. The authors have also responded to the vast majority of reviewer comments (see below). Further review comments are made as two contributions: 1) A summary of the updated manuscript below and; 2) Remarks to the author responses on reviewer comments in italics and blue when applicable (*no additional response was made when the authors response was considered satisfactory*).

Overall I find the manuscript to be of significant interest, but further revisions are recommended as outlined below:

Abstract line 41 – the second part of this sentence could be clearer to illustrate how the methodology relies on a change in protein turnover when chaperones such as HSP90 are inhibited.

We have updated the Abstract in the main text (lines 42-44) to clarify this point, thank you.

Introduction row 61 – The limitation raised by the authors may be applicable to membrane-bound proteins, but is hardly a specific limitation of CETSA as cell lysis is readily achieved in CETSA and other experiments. The use of immunofluorescence as applied by the authors is similarly dependent on sufficient permeabilization and even staining (the corresponding use of immunofluorescence was also recently demonstrated for CETSA; Axelsson et al. 2018 ACS Chemical Biology). Similarly on row 65 the authors indicate that several techniques rely on “ectopic” expression of tagged protein, which is not necessarily true as it depends on the means for expression of tagged protein. The value of the new technology can be perceived without unnecessary deterioration of the value of alternative techniques

We agree with the reviewer's points and believe that our method aims to be complementary to existing techniques. We have changed the text in the Introduction (lines 54-67) hopefully sufficiently to address these concerns and we discuss the requirements and applications of the different techniques, thank you.

Results row 108 – the measured EC₅₀ should be compared with literature expectations.

We have added in text discussing our data in relation to the published cellular EC₅₀ values for endoxifen (lines 109-110).

Results row 120 – this is what forms the basis of the dynamic proteomics approach, consider citing appropriate literature (see below)

Thank you for the suggestion. We have inserted a reference to the dynamic proteomics approach and cited Cohen et al. on line 123 (as well as in the introduction as one of the current methods to detect drug-target engagement, line 66-67).

Results row 139 - the measured EC₅₀ values and their ranking should be compared with literature expectations

We have added in text discussing our data in relation to the published cellular and biochemical IC₅₀s for the 3 HER2 inhibitors where available and to our own biochemical IC₅₀ data shown in Supplemental Table 1 (lines 144-149).

Figure 4F-H – It is not clear from the figure legends or the materials and methods what the data are normalized to in these figures. If it is towards DMSO controls why does it start at appr. 12 and 15 at the two different compound concentrations?

We have more explicitly described the normalization method in the legend for Figure 4 (lines 718-719), in the methods section (lines 483-486) and in the figure legends for Supplementary Figures 5, and 6. Individual data points are calculated as the percentage of sum total TMT signal for the protein across all treatment groups. This means that the sum of all normalized individual values across treatment groups for any one protein will be 100. This method enables the normalization of each individual measurement to an estimate of protein abundance derived from the largest possible sample size, in order to provide a measure of relative changes in the level of each individual protein. We also corrected the relative abundance plots for ER α and c-RAF in Supplementary Figure 5A and B, which were not normalized correctly as described; and we have standardized the method used to calculate statistical significance for log₂ratio values (shown in Volcano plots) and for relative abundance plots (We had inadvertently been using two different statistical methods for these two forms of data; all methods now apply 2-tailed tests). This slightly altered our assessment of the significance of data for a small number of proteins.

Results row 207 – A better mechanistic explanation of observed stabilization is needed. For independent confirmation that TAK-285 binds PHLDA2 the authors make use of lysate CETSA (Supplementary Figure 11), which demonstrates a destabilization in the presence of 10 μ M compound. The simplest explanation for destabilization, especially in a dilute cell lysate where prestabilization by endogenous metabolites is diminished, is that the compound binds preferentially to the denatured protein, thus driving the thermal aggregation temperature to lower values. Such destabilization agrees with the finding of increased protein turnover to yield reduced protein levels in the presence of increasing TAK-285 concentrations using “dynamic proteomics” (Supplementary Figure 10 A and C). The latter change is however not as pronounced in the MS data (Figure 4H), can the authors comment on the reason for this?

Also here I cannot follow why the normalization of protein levels is done differently for Figure 4H (where the DMSO controls start at a relative abundance of 12). As identified by the authors treatment with 17-AAG results in reduced cellular turnover of PHLDA1/2, but a key outstanding question is what protein form the compound binds to, i.e. folded or unfolded? It would be useful to complement these studies with a binding experiment towards isolated folded protein, e.g. a thermal shift experiment using DSF, to understand the underlying mechanism of action in detail.

We are happy to expand our discussion of the potential mechanism of stabilization and destabilization of PHLDA2 by TAK-285. We have added this into the discussion section (lines 319-325). The HIPStA quantitative fluorescence imaging data (now Supplemental Figure 12) suggest that there are two populations of PHLDA2; in the presence of active HSP90 chaperone approximately 50% of PHLDA2 is in a form that is destabilized by TAK-285, while in the absence of HSP90 chaperone function approximately 50% of PHLDA2 is in a form that is stabilized by TAK-285. In the cell lysate CETSA study (now Supplementary Figure 13) only a destabilizing effect was observed, suggesting that dilution of associating cofactors or accessory proteins might modulate the effect that TAK-285 has on PHLDA2 stability.

Regarding the potential reasons for why the degree of reduction in PHLDA2 proteins levels caused by TAK-285 differ between the MS data in Figure 4H and the HIPStA quantitative fluorescence imaging data (now shown in Supplementary Figure 12A and C): We can speculate that the two methods may differ in their ability to detect partially cleaved forms of PHLDA2, or covalently modified forms of the protein, which may result in the observations. The two peptides used to detect PHLDA2 in the mass spec study are both in the N-terminus of the protein, while at least one of the antibodies used to detect PHLDA2 in the quantitative fluorescence imaging study recognizes an epitope in the C-terminus (the exact epitope of the second antibody is not known). Partial cleavage of the protein may result in a greater loss of signal by one method versus the other. Alternatively, ubiquitination of the PHLDA2 might differentially affect the ability of the two techniques to detect protein. The PHLDA2 C-terminal epitope detected by one of the antibodies and one of the peptides used to detect PHLDA2 in the MS study, contain lysine residues that might be sites for ubiquitination during protein destabilization and induced degradation. There may also be a more trivial explanation of the differences in the extent of change in PHLDA2 protein detected by the two experimental methods that is related to the relative cell densities, assay volumes and the partitioning of the compound in the cell layer. In our experience such factors can influence the apparent potency of drug effects in cell assays. The cell density and amount of cell material (relative to cell culture medium) was significantly higher in the global proteomics profiling studies than in the quantitative fluorescence imaging studies.

Regarding the question of the method of normalization in the relative abundance values – see our earlier comments and the steps we have taken to hopefully make this clearer. Regarding the specific comments about Figure 4H - The values in Figure 4H are normalized in the same manner as the other data sets. The normalization is to the sum total of the protein detected across the different treatment sets for that protein.

Regarding the question of “what protein form the compound binds to, i.e. folded or unfolded?” - Although we believe it is out of scope of the manuscript to investigate which conformations or sub-populations of PHLDA2 bind TAK-285, we attempted a DSF study, as suggested, using purchased recombinant PHLDA2 (ProSci Catalog# 92-098). Unfortunately the protein did not yield good quality thermal melt curves; and in addition a confounding technical factor was that the presence of TAK-285 by itself appeared to bind the reporter dye (SyproOrange) and increase fluorescence. We were thus unable to obtain good quality DSF data and could not test whether this method could

answer the question posed by the reviewer.

Discussion row 273 – this is where a comparison between HIPStA and MS CETSA would benefit from a more comprehensive comparison of the overlap and mismatches

We have included a more comprehensive comparison of the overlap and mismatches between the HIPStA and CETSA global proteomics profiling in the results section (lines 203-220), added Supplementary Figures 8 and 9 and Supplementary tables 3 and 4, and more discussion of potential reasons for these differences at the point suggested by the reviewer (lines 304 – 309).

Point-by-point response to reviewer comments:

No additional response was made when the authors response was considered satisfactory.

Reviewer #1

As example, the authors chose HER2 inhibition by known HER2 inhibitors (Lapatinib, HY-14674 and TAK-285) in the MCF-7 breast cancer cell line. Using fluorescence readout, they also targeted two other classes of HSP90 clients, namely nuclear hormone receptors and cytoplasmic kinases. I suggest moving these results to a main figure because it shows broader application of the assay.

We agree with the Reviewer and have moved the data on ER α and cRAF to the main figures to show generalizability to other HSP90 clients.

For the mass spectrometry experiment, the authors only chose the fairly selective, ATP competitive TAK-285 inhibitor. How does this assay coupled with MS readout perform for other inhibitors especially unselective kinase inhibitors like staurosporine or dasatinib? This might be a better example to investigate the overall performance of HIPStA. Does one recover most of the known targets?

To address this, we have performed an additional mass spectrometry experiment using staurosporine. From our results, we observe elevated protein levels of 39 kinases and observe the most significant HIPStA response in 4 kinases (MAP3K20 (ZAK), ERBB2, STK32C and MASTL (GWL)). For other proteins, we observe results that are consistent with some, but not all, observations from previous CETSA studies (Supp. Figure 8 in revised manuscript).

This makes an excellent contribution to the manuscript, although the comparison with CETSA results should be further improved. CETSA is known to have blind spots when there is no

thermal shift despite a physical interaction (hence I would not expect perfect consistency). It will however be useful for the reader to have the correlation described in more detail for the four best responding kinases (see below for further comments and suggestions on a supplementary table).

Thank you for the suggestion of adding a supplementary table to help show this comparison. We have added Supplementary Tables 3 and 4 to show how the HIPStA hits fared in the CETSA study and how the CETSA hits fared in the HIPStA study. We have also added a new Supplementary Figure 8 to show the comparison between the two studies regarding the number of proteins being detected and being classified as “hits”, in Venn Diagram format; we have added a new Supplementary Figure 9 to show how the various filters applied to the HIPStA study whittles down the corresponding hits in the CETSA study. We have added more extensive description of our results in relation to CETSA data (lines 195 to 220) and a more extensive discussion (lines 300 to 309) of this comparison. Altogether we believe these additional analyses of the HIPStA and CETSA data address the reviewers’ concerns and improve the manuscript.

Besides HER2, they identified two previously unknown, non-kinase off targets, PHLDA1 and PHLDA2. However, comparing the MS results to the selectivity profiling screen, Yes-kinase, which shows 86% inhibition is detected by MS but not identified as a potential target by the HIPStA assay. Does this imply the assay only recovers very strong binders? What are the detection limits of the assay and which affinities can be measured?

The reviewer raises the question of whether the fact that we did not detect the interaction of TAK-285 with Yes kinase (a biochemical target of TAK-285) suggests that the technique will only detect very strong binders. To address this we applied the alternative target engagement method, nanoBRET, using nanoluciferase tagged Yes kinase and an active site tracer. Our data indicate that the nanoBRET method was also unable to detect the binding of TAK-285 to Yes kinase, in contrast to our positive control dasatinib (Supp. Fig. 7 in revised manuscript). These results support the conclusions of the HIPStA study, that TAK-285 does not interact with Yes kinase in the cellular context. However we should also comment that, from our MS results, we observed that the levels of Yes kinase did not change drastically with 17-AAG treatment alone (Fig. 4B and Supp. Fig. 6A-C in revised manuscript). This would result in a diminished possible dynamic range for rescue by TAK-285, compared to HER2.

Does binding affinity correlate with degradation rate of HSP90?

While we have not investigated degradation of HSP90 directly, an increased degradation rate of HSP90 would likely have the same effect as increasing 17-AAG, since both result in a decrease of functional HSP90. In our case, since we are using the same cells within each experiment and assume the same rate of protein degradation, we attribute the change in our measurements to 17-AAG’s inhibition of HSP90’s chaperone function.

This question relates to a question from reviewer 2 on the measured XC_{50} values and how these correlate with measurements from other techniques (see below).

Thank you for the clarification. We have added in text discussing our data in relation to the published cellular EC_{50} values for endoxifen (lines 109-110). We have added in text discussing our data in relation to the published cellular and biochemical IC_{50} s for the 3 HER2 inhibitors where available and to our own biochemical IC_{50} data shown in Supplemental Table 1 (lines 144-149).

How does this assay perform compared to other (kinase) binding and activity assays like CETSA, multiplexed inhibitor beads/kinobeads, KinomeScan?

To address this question of how HIPStA compares to other binding techniques, we carried out CETSA studies of the interaction of TAK-285 with PHLDA2, which was initially identified as a potential TAK-285 interacting protein in the HIPStA studies. Our data corroborate the observations in the HIPStA study, that TAK-285 modulates PHLDA2 stability (Supp. Figure 11 in revised manuscript). In an additional staurosporine HIPStA global proteomics profiling experiment (Supp. Figure 8 in revised manuscript), we identify several proteins that were also identified in a previously published CETSA study (but not all), as well as several additional potential targets of staurosporine. We also confirm that TAK-285 does not bind to YES kinase (as suggested by HIPStA) using a nanoBRET based binding assay.

See previous comment on the need to improve the (quantitative) comparison between techniques. As the techniques rely on different underlying principles excellent agreement is not expected. However, when a protein has shown a stabilizing response in a CETSA experiment AND is demonstrated as a HSP90 substrate there would be an expectation of agreement as it is then amenable to both approaches. It would be useful to see a comparison for this selected subset of proteins (out of the 18 and 27 at >25% at the different timepoints), e.g. in a supplementary table. Are there examples of these that respond to staurosporine in CETSA, but not in HIPStA?

See our earlier response to a similar suggestion. We have added new Supplementary Figures 8 (Venn diagram) and 9 (CETSA vs HIPStA scatter plots and volcano plots); we have added Supplemental Tables 3 and 4 showing HIPStA vs CETSA comparisons. We have added more extensive description of our results in relation to CETSA data (lines 195 to 220) and a more extensive discussion (lines 300 to 309) of this comparison.

Please update Supp Fig 5, the figure legend and volcano plot axes and titles do not match.

We have fixed this and have changed the axes and titles to be consistent with the figure legends, thank you.

In Fig 2DE, also mark other TAK285 targets identified in the recombinant kinase panel assay.

We have updated Fig. 2BDE (Fig. 4 in revised manuscript) to include Lck, Yes, Fyn, Fgr, PKD1, PKD2, and AurKB (given they were detected in the respective experiment).

Supp Fig 3. Panels E, F are duplicated.

The Panel E and F of Supp. Fig. 3 (Fig. 2 in revised manuscript) show c-RAF levels when cells are treated with either the BGB283 (enantiomer) compound (E) or the Novartis pan-RAF inhibitor (F) and represent two different sets of experiments/results.

I suggest moving spectrum in 2 F into supplement.

We have moved the spectrum in Fig. 2F (Fig. 4F in revised manuscript) to the supplement, thank you for the suggestion.

Reviewer #2

The observations made here would have to be shown also with other drugs, preferably those with a broad and well characterized target profile to deduce how widely applicable this method is. This will be informative particularly considering that only target proteins that are chaperoned by HSP90 will be captured by Hipsta, as the authors somewhat indicate correctly.

In a similar response to the previous reviewer's concern, we have performed an global proteomics profiling experiment using staurosporine. We observe elevated protein levels of 39 kinases and observe the most significant HIPStA response in 4 kinases (MAP3K20 (ZAK), ERBB2, STK32C and MASTL (GWL)). For other proteins, we observe results that are consistent with some, but not all, observations from previous CETSA studies (Supp. Figure 8 in revised manuscript) (Supp. Figure 8 in revised manuscript).

The cell line used in this study appears to be engineered to overexpress HER2. It would be important to demonstrate that Hipsta works equally well with endogenous protein using for instance an HER2 amplified cell line like SKBR3 or entirely different cells and targets. The authors show that Hipsta works also for Era and RAF1, but this is only shown by IF. It is not clear, if MS detection, which is necessary for unbiased identification of targets, would work equally well.

We have repeated our assay using TAK-285 in two other cell lines SK-BR-3, as suggested by the reviewer, and AU565. We observe that TAK-285, but not dacomitinib, protects against 17-AAG induced degradation of HER2, consistent with our observations in the MCF7 neoHER2 line (Supp. Figure 3 in revised manuscript). Furthermore, our new mass spectrometry data using staurosporine also addresses this point as we detected HIPStA responses for a range of proteins using the non-selective kinase inhibitor. This data suggests that HIPStA is sensitive enough to identify endogenous targets.

It is important to clearly state the strengths as well as the limitations of any new method. Thus it is important to state that Hipsta will only cover target proteins that are stabilized by HSP90 chaperoning although a drug may have multiple targets that do not depend on HSP90. In addition, the authors should discuss that with the current experimental design they won't be able to determine an IC50 from a mass spectrometry based experiment and would therefore require independent downstream validation of new targets. What is the IC50 value for EGFR as determined by Hipsta?

We have included in our discussion that HIPStA in its current state is limited to HSP90 clients, but that it could also potentially be adapted to surveying clients of other chaperones. We present HIPStA paired with mass spectrometry as a strategy for identifying targets of a drug in an unbiased manner. However, we also show using multiple approaches, as in the case of PHLDA2, that additional biochemical characterization is required for validation of identified targets.

This reply is incomplete as it does not respond to the question of how the IC50 values derived from HIPStA relates to measurements using other approaches. A comparison with literature values for a couple of examples based on already existing data would be useful to set expectations correctly.

We have added in text discussing our ER α data in relation to the published cellular EC50 values for endoxifen (lines 109-110). We have added in text discussing our HER2 data in relation to the

published cellular and biochemical IC50s for the 3 HER2 inhibitors where available and to our own biochemical IC50 data shown in Supplementary Table 1 (lines 144-149). Regarding the question of: “What is the IC50 value for EGFR as determined by Hipsta?” – We have re-emphasized on line 181 that the detection of EGFR in the HIPStA-MS was ambiguous because the peptides used to detect it were also present in ERBB2 and that EGFR expression was not confirmed by immunofluorescent staining; we therefore cannot derive a HIPStA IC50 for EGFR from the cells used in this study.

Where in Figure 2 are ERBB4 and LCK? Are other known targets like PKD1/2 or YES observed by mass spec?

We have updated Fig. 2BDE (Fig. 4 in revised manuscript) to include YES, FYN, FGR, PRKD1 (KPCD1), PRKD2 (KPCD2), and AURKB (given they were detected in the respective experiment). While these proteins were detected by mass spectrometry, we did not observe a decrease by 17-AAG that was rescued by TAK-285. ERBB4 and LCK were not detected in these experiments.

The concentration range queried for the BGB-283 compound is inadequate as it appears to be not potent enough for RAF1 and does barely protect RAF1 at 1 μ M. Higher concentrations would need to be tested.

For BGB-283 (Enant.), we tested concentrations up to 5 μ M. We observe a protection effect as shown by the elevation of the curve plateau (Figure 2G in revised manuscript). We do note that at high concentrations of BGB-283 (Enant.), we observe an effect on increased c-RAF levels (Figure 2E in revised manuscript), which can confound HIPStA measurements.

As most of the stabilization/destabilization effects observed are somewhat moderate, Figure 2 would be more informative if the x axes were rescaled with finer graduation.

To maintain figure clarity, we did not add finer gradation to the x-axis. However, if a reader wishes to more carefully analyze the data, the entries with significant responses for the mass spectrometry experiment using TAK-285 can be found in Supplementary Table 2.

Several supplementary figure (S4, S5C/D, S6A/B) are not discussed in the text although their inclusion does make sense and adds value to the manuscript.

We have included discussion of these supplementary figures in the main text to support our claims, thank you.

Line 77: This refers to Figure 1E, not 2E.

We have corrected this figure reference.

Line 64: It would be more appropriate to talk of a half-life greater than 500 min instead of 511 as the experiment is ended at 500 min.

We have changed the half-life to >500 minutes, as suggested, thank you.

The double bond in Figure S2C needs to be redrawn.

We have included a new structure for Endoxifen in Figure S2C (Fig. 1C in revised manuscript).

The purpose of Figure S7 should be mentioned in the manuscript text.

We have included reference to this figure (Supp. Fig. 9 in revised manuscript) to show confidence in our assignments of each of these proteins, due to unique peptides between each.

Line 114: Please synchronize the nomenclature on PHLA1/2.

We have changed the manuscript to refer to the proteins as PHLDA1/2 consistently.

Line 114: This refers to Figure 2D-F, not 4D-F

We have corrected this figure reference, thank you.

Line 210: TMT 10 plex or 11plex?

Our samples were prepared with the TMT10plex labeling reagent + TMT11 reagent set from Thermo. We have updated the methods to clarify this.

in Figure 2: Does the dashed horizontal line indicate a p value of 0.05?

In Figure 2 (Figure 4 of revised manuscript), the y-axis is the $-\log(p\text{-value})$, so the bottom dashed line indicates a p-value of 0.05 and the top dashed line indicates a p-value of 0.01.

Reviewer #3

The manuscript is immature and lacks appropriate subheadings to facilitate reading. For example, the abstract is very short and lack details on the results or the potential future application of the methodology. Furthermore, the background description on alternative methodologies is limited to a few examples and in some cases incorrect. To exemplify: CETSA has clear limitations but the description of a complex mixture of concentrations and temperatures is not correct. There is ample literature showing how it can be readily applied for both screen purposes and for full concentration response profiling (see e.g. the recent publication from Shaw et al. in SLAS Discovery and last years publication from NCATS NIH on 17 model systems besides the cited literature). The concerns with CETSA are rather related to the need to heat cell samples, resulting in potential impacts on cell permeability and alterations of established equilibria. Other techniques for cellular target engagement, such as NanoBRET, FLIM, and dynamic proteomics (InCell Hunter), are left out. The lack of citations on the literature on dynamic proteomics is difficult to understand, I feel it should be mentioned at is relies on the same underlying concept, i.e. ligand induced alterations in cellular protein turnover. The authors should consider extending the background description and include these related approaches to put their work in perspective.

We have expanded our Abstract, included appropriate subheadings in our Results, and have included a Discussion section. We have also extended our Background section to include further adaptations of CETSA as well as other techniques that the Reviewer has mentioned. We include the overall limitations of these other approaches and how HIPStA overcomes some of these

issues. Thank you for the feedback.

This background description is much improved, but relevant references on the closely related dynamic proteomics approach are still missing. An early work on the application of altered cellular turnover in response to compound treatment is Cohen et al. in Science 2008 (Dynamic proteomics of individual cancer cells in response to a drug).

We have inserted a reference to the dynamic proteomics approach and cited Cohen et al. on line 123 (as well as in the introduction as one of the current methods to detect drug-target engagement, line 66-67).

The authors claim that prior titration of a suitable 17-AAG concentration results in an assay with significantly improved throughput (page 3, line 74-75). It appears such titration is analogous the thermal aggregation curve that is established prior to HT CETSA experiments (to define a single temperature for profiling purposes), the titration of a fluorescent tracer concentration in NanoBRET experiments for use in competition experiments and a suitable endpoint time in dynamic proteomics (InCell Hunter) experiments. All these then allow characterization of full concentration responses so any throughput advantages should be more clearly described if the argument remains. The methodology is clearly complementary and may be responsive in cases where dynamic proteomics is not, but I am unable to understand the throughput argument.

We agree that alternative methods can also be adapted to have high throughput. We have adjusted our text to include this. While HIPStA offers the advantage of being a high-throughput technique, we no longer claim that this is unique to HIPStA, and we agree that HIPStA is complementary to the other techniques. We have included this in our Background and Discussion sections.

A proteomics experiment is then used to examine to what extent the methodology can be applied for identification of target proteins in a cellular setting. However, this experiment is based on a single HSP90 inhibitor. The authors should consider validating their findings using additional inhibitors or other means to suppress HSP90 activity. This is especially important as 17-AAG is also applied at a single concentration (1 μ M) for the proteomics experiments and used to support arguments that what is identified is the HSP90 client population (page 4, row 94). This claim needs to be supported by more rigorous comparisons with previous literature (the volcano plots in supplementary materials do not clearly demonstrate the correlation between herein achieved results and previous literature. Some support comes from the observation of kinase destabilization (broadly), but also this comparison needs to be on a more detailed level to understand correlations with prior data (or alternatively measured using an orthogonal approach).

We based our HIPStA assay development on 17-AAG since it is widely available to researchers, and the geldanamycin analogs, such as 17-AAG, are probably the best characterized of the HSP90 inhibitors, inhibiting all four mammalian isoforms of HSP90. While there are some off-target effects known for geldanamycin analogs, they are well characterized (unlike many other HSP90 inhibitors) and typically become apparent at higher concentrations (>10 μ M) than used in this study (Neckers et al Cell Stress and Chaperones (2018) 23:467–482). However, to address the question of whether another HSP90 inhibitor could be used instead of 17-AAG, we carried out a HIPStA study of HER2 stabilization by TAK-285 in SK-BR-3 cells, using the tropanone-based HSP90 inhibitor XL888; the results are essentially the same as those for 17-AAG (Supp. Fig. 3 in revised manuscript). Also, in a similar response to the previous reviewers' concerns, we

have performed an additional mass spectrometry experiment using the non-selective inhibitor staurosporine. We observe elevated protein levels of 39 kinases and observe the most significant HIPStA response in 4 kinases (MAP3K20 (ZAK), ERBB2, STK32C and MASTL (GWL)). For other proteins, we observe results that are consistent with some, but not all, observations from previous CETSA studies (Supp. Figure 8 in revised manuscript).

Similarly, biochemical data highlight the interaction of TAK-285 with seven kinases, but the only example that is highlighted as a hit in the proteomics experiment is HER2, which coincides with what was used to first validate the approach. Although it is fully understandable that not all seven kinases are targeted in a cellular setting, the authors should make this comparison, clearly illustrate the data and allow the readers to form their own opinion.

As similarly requested by previous reviewers, we have updated Fig. 2BDE (Fig. 4 in revised manuscript) to include Yes, Fyn, Fgr, PRKD1 (KPCD1), PRKD2 (KPCD2), and AURKB (given they were detected in the respective experiment).

Along the same lines the identification of PHLDA1/PHLDA2 as potential binders of TAK-285 in a cellular setting is interesting. However, the confirmation of this observation is only achieved by means of changing detection methodology, i.e. from MS to immunofluorescence. Given that this is a new approach, it would be useful to validate these findings using e.g. biochemical assays. Certainly, to achieve the standard of Nature Communications, several different approaches to validate the outcome can be expected. Possibilities besides biochemical assays on recombinant proteins include for example CETSA or resistance to proteolysis in cell lysates (DARTS).

We have further validated our findings from the TAK-285 study using CETSA to assess the effect of TAK-285 on the thermal stability of PHLDA2 in cell lysates (Supp. Fig. 11 in revised manuscript). Consistent with our mass spectrometry results and immunofluorescence results, the CETSA data indicate that TAK-285 modulates PHLDA2 stability, although it is interesting to note that in the absence of HSP90 inhibition, TAK-285 destabilizes PHLDA2 in both CETSA and HIPStA studies.

The discussion is missing.

We have expanded our manuscript considerably and it now includes a discussion section, with key points that reviewers have addressed above, thank you.